# Multigene phylogeny and morphological descriptions of five species of *Agaricus* sect. *Minores* from subtropical climate zones of Pakistan

**Hira Bashir**[1,2¤]*, **Muhammad Asif**[1], **Aneeqa Ghafoor**[1], **Abdul Rehman Niazi**[1], **Abdul Nasir Khalid**[1], **Gulnaz Parveen**[3], **Nidaa Harun**[4], **Najam-ul-Sehar Afshan**[1], **Ayesha Bibi**[1], **Philippe Callac**[2]

**1** Fungal Biology and Systematics Laboratory, Institute of Botany, University of the Punjab, Quaid- e-Azam Campus, Lahore, Pakistan, **2** INRAE, MycSA, CS 20032, 33882 Villenave d'Ornon, France, **3** Department of Botany, Women University Swabi, Swabi, Pakistan, **4** Department of Botany, University of Okara, Okara, Pakistan

¤ Current address: Department of Botany, Women University Mardan, Khyber Pakhtunkhwa, Pakistan
* drhirabashir@wumardan.edu.pk

**Data Availability Statement:** https://www.treebase.org/treebase-web/home.html;jsessionid=1F4ADD78BB9C492CEC9BE51AE74A7115 https://

## Abstract

The genus *Agaricus* includes more than 500 species mostly containing the edible and cultivated species worldwide. As part of the ongoing studies on the biodiversity of genus *Agaricus* in Pakistan, our objective was to focus on *A.* sect. *Minores* which is the largest section of the genus. In the first phylogenetic analyses based on the ITS region of the nuclear ribosomal DNA, our sample included specimens of 97 named species, 27 unnamed species, and 31 specimens (29 newly generated sequences in this study) from subtropical climate zones of Pakistan that likely belong to this section based on their morphology. The 31 specimens grouped into five distinct, well-supported clades corresponding to five species: *A. glabriusculus* already known from Pakistan and India, *A. robustulus* first recorded from Pakistan and briefly described here but already known from Bénin, Malaysia, China, and Thailand, and three possibly endemic new species described in detail *A. badiosquamulosus* sp. nov., *A. dunensis* sp. nov., and *A. violaceopunctatus* sp. nov. The sixth species currently known in Pakistan, including *A. latiumbonatus* also found in Thailand, were included in a multigene tree based on ITS, LSU, and Tef-1α sequence data. They all belong to a large pantropical paraphyletic group while most temperate species belong to a distinct clade, which includes about half of the species of the section. The current study aims to propose three novel species of genus *Agaricus* based on comprehensive morphological as well as molecular phylogenetic evidences from Pakistan.

## Introduction

The genus *Agaricus* (*Basidiomycota*) is the type genus of *Agaricaceae* containing saprophytic species. The species are adapted to live in various climatic regions as subtropical, tropical,

www.mycobank.org/ https://www.ncbi.nlm.nih.gov/nucleotide/ All the accession numbers have already been mentioned in the manuscript.

**Funding:** The authors received no specific funding for this work.

**Competing interests:** The authors have declared that no competing interests exist.

temperate, Caribbean islands and deserts. Several species are consumed and cultivated because of their nutritional and medicinal importance such as *A. bisporus* (J.E. Lange) Imbach, *A. arvensis* Schaeff. and *A. subrufescens* Peck [1, 2–4].

The classification of the genus is mainly proposed by Zhao et al. [5] which was revised using molecular clock methods [6]. Five monophyletic clades A, B, C, D, and E were proposed having stem ages ranging from 30–33 My ago. Each clade is considered as subgenus. Amongst these five subgenera, Clade A is considered as a subgenus *Minores* [6]. It includes mainly the section *Minores* reported previously by Zhao et al. [5] that diverged 30 My ago and is subdivided into Clade A1/ sect. *Minores*, Clade A2/sect. unknown or sect. *Laeticolores* and sect. *Laeticolores*. *Agaricus* sect. *Laeticolores* was represented by only one species *A. rufoaurantiacus* (LAPAM15) [6] which was reported as subgen. *Minoriopsis* by Chen et al. [7]. In the same study by Chen et al. [7] another new section *Leucocarpi* has been introduced and Clade A2/ sect. 1 still remains unnamed. The classification of subgen. *Minores* is recently revised where subgen. *Minoriopsis* is splitted into *A.* sect. *Kerrigania* and *Minoriopsis*. *Agaricus* sect. *Minores* is sub-divided into three sections viz. *A.* sect. *Pantropicales* L.A. Parra, Angelini, B. Ortiz, Linda J. Chen & Callac, *A.* sect. *Minores* (Fr.) Henn., and *A.* sect. *Leucocarpi* Linda J. Chen & Callac [6–8]. Later, He et al. [9] has introduced a new section of *Agaricus* subg. *Pseudochitonia* and now the number of sections are increased to 24. According to Index Fungorum data, there are about 6000 species of *Agricus* recorded from all over the world so far (accessed in April 2022).

Generally the species of *A.* sect. *Minores* are distributed world wide in tropical and temperate areas [5, 10] and some are found in extreme climates like secotioid species, adapted well to desert conditions [11, 12]. However, the newly reported species in this study were collected mainly from subtropical areas of Pakistan and the desert but none of them is sectioned. Previously 35 species of the genus *Agaricus* have been reported from Pakistan [7, 13–20] amongst these only two species belong to *Agaricus* sect. *Minores* [17]. Recently, two new species viz. *A. midnapurensis* and *A. purpureosquamulosus* and a new record of *A. glabriusculus* have been documented using ITS region from India [21] and *Agaricus parviumbrus* is documented as a new species in Fungal Planet description sheets [22].

The data of all the species in this study is a significant contribution to the knowledge of genus *Agaricus* from Pakistan. A few studies have been conducted based on multigene and morphological data so far. The heteromorphic sites of ITS region within and between the closely related taxa are analyzed in this study.

## Materials & methods

### Specimen collection and sampling sites description

A total of 29 specimens of genus *Agaricus* were collected from different semi-arid and arid localities of Punjab province, Pakistan. Based on morphological observations and fresh basidiomata field notes, the specimens likely belonged to *Agaricus* sect. *Minores*. As part of the current study on the genus, surveys were conducted in different subtropical areas of Punjab province i.e., Lahore; Toba Tek Singh; Haroonabad, district Bahawalnagar and Lal Suhanra National Park, Cholistan desert, district Bahawalpur, Pakistan.

Lal Suhanra National Park lies in southeastern Punjab, Pakistan. This national park is part of the Cholistan desert having fungal flora that has not yet been studied so far. It is a subtropical region characterized by very low annual rainfall (90–200 mm) in monsoon season, low relative humidity (60%) and high temperature (50°C) in summer [23]. It is located 32 km away from district Bahawalpur.

District Bahawalpur lies in southern side of Punjab, Pakistan. This subtropical region receives average annual rainfall fluctuating from 90 to 200 mm only [24]. The average annual temperature is more or less similar to Lal Suhanra National Park ranging from 6 to 10˚C in winters and 49 to 52˚C in summers [25].

The other collection site lies in Haroonabad, District Bahawalnagar, Punjab, Pakistan adjacent to Cholistan desert which is characterized by hot and dry climate with an extreme temperature of 50˚C and minimum temperature of 11˚C. The average annual rainfall is just 99 mm [26, 27]. The basidiomata (BWN 34, BWN 67, BWN 85) were collected along the bank of three R (3R) irrigation canal under *Acacia nilotica* from the loamy soil.

Toba Tek Singh is a district of Punjab with a subtropical climate. The maximum temperature is recorded at 40.7˚C and a minimum of 6˚C with an average annual rainfall of 254–381 mm [28].

Changa Manga Forest lies in the southeast of the district Lahore. This is a subtropical area with minimum to maximum temperature ranges from 5.9 to 39.6˚C, respectively, and the annual rainfall is about 650 mm [29].

## Morphological characterization

The fresh specimens were photographed during field surveys at collection sites. Collected basidiomata were carefully tagged and dried in sunlight or by using a fan heater. All the dried specimens were packed in zipper bags and brought to the laboratory for further processing. Morphological observations such as color, size, and shape of pileus, stipe, and lamellae were noted [30]. Other fresh sporocarp characters were also observed and noted during collection survey including discoloration upon bruising/handling, odor, and different chemical reactions such as KOH and Schaeffer's reaction [31]. Color codes were given from Munsell's Soil Color Charts [32].

Tissues from different parts of fruiting bodies were rehydrated in 5% KOH (w/v) solution followed by staining in 1% aqueous Congo red (w/v). Anatomical characeterization included size and shape of basidia, basidiospores, cheilocystidia, pileipellis and stipitipellis and their measurements were taken using a light microscope at 100X (MX4300H, Meiji Techo Co., Ltd., Japan). All the microscopic features were represented by at least 20 measurements each. The dimensions of basidiospores are provided as (a) b–c (d) × (e) f–g (h), [avX, Qm, n = i × j] where b–c and, f–g includes the spore length and width, respectively, between the 5th percentile and the 95th percentile, (a) and (d) the minimum or maximum values of spore lengths recorded, similarly (e) and (h) the minimum or maximum values of basidiospores width were recorded, avX the mean of length by width ± SD (standard deviation), Qm the mean of Q coefficient (length/width ratio), n represents the total number (i) of basidiospores measurements of each sample/collection and (j) indicates number of total samples/collections measured. Measurements of other anatomical structures as basidia, cheilocystidia, hyphal structures including pileipellis, and stipitipellis were available in dried samples included the range between the minimum or maximum values calculated in length and width. The measurement of basidia was considered excluding sterigmata. Short descriptions of all the new species have been deposited in MycoBank.

**Phylogenetic observations.** For ITS phylogenetic analyses, a dataset of 158 specimens belonging to 95 nammed and 27 unammed species within *A*. sect. *Minores* was used. The remaining taxa belong to *A*. sect. *Leucocarpi* (01), *A*. sect. *Pantropicales* (03) and *A*. subg. *Minoriopsis* (02). The sequences for phylogenetic analyses were selected from initial BLAST results and in previously referenced studies on *A*. sect. *Minores* [17, 31].

For multigene phylogenetic analyses, dataset of 100 specimens with characters was included belonging to 72 named and 26 unnamed species including *A*. sect. *Leucocarpi* (01), *A*. sect. *Pantropicales* (03), Incertae sedis (02), and 02 from *A*. sect. *Agaricus* as outgroup taxa.

**DNA extraction, PCR amplification and DNA sequencing.**   DNA was extracted from the dried specimens using the 2% CTAB method with some modifications as described by Zhao et al. 2011 [5]. PCR amplification was performed of three nrDNA regions viz. ITS (Internal Transcribed Spacer region) using ITS1F/ITS4 primers [33, 34] LSU (large subunit fragment) using LROR/LR5 set of primers [35] and EF1 (translation elongation factor) utilizing EF1-983F/EF1-1567R combination of primers [36]. In PCR cycle, denaturation was done at 95˚C followed by annealing of 35 cycles at 94˚C and final extension at 72˚C. The PCR products were sequenced from BGI (Beijing Genomic Institute) and Genewiz, UK. All the newly generated sequences during this study have been submitted to GenBank.

**DNA sequence alignments.**   Sequence alignment for ITS and multigene datasets was performed independently, using T-cofee version 8.99 [37] then final adjustment was done manually in Bioedit version 7.2.0. The CIPRES PORTAL v. 3.1 [38] was used for maximum likelihood (ML) analyses. Both ITS and multigene phylogeentic analyses were conducted for each alignment separately using RAXML–HPC2 v 8.1 [39]. Topology of phylogram was retrieved at 1000 bootstrap replicates. Significant support was considered to be BS≥50. No significant inconsistency between the ITS and multigene sequence data was observed, therefore ITS, LSU, and TEF1 sequences were integrated in BioEdit for final phylogenetic analyses. The multigene data was segmented into ITS, LSU, TEF1 (intron and coding sites).

**Data analyses.**   The most suitable substitution model for each region was inferred with MrModeltest version 2.2 [40]: SYM+G for all partitioned DNA regions. Bayesian Inference (BI) [41] was completed with MrBayes v. 3.1.2. Six Markov chains were run for one million generations and sampled every 100th generation. Burn-in was indicated in TRACER v. 1.6 [41] from trace plots of likelihood and finally discarded. The resulting phylograms were opened in FigTree version 1.4.2 (http://tree.bio.ed.ac.uk/sofware/fgtree/). The final alignments of both ITS and multigene dataset were deposited in TreeBASE (ID = 30723).

**Species-specific ITS markers.**   Taxa, clades, sub-sections, or sections characterizing markers were located in the final alignment of 153 total ITS sequences of *A*. sect. *Minores*. These markers are provided with capital letters and are shown along with flanking sequences. IUPAC codes, for example Y, do not indicate ambiguous nucleotide (C or T) but heteromorphic site (C and T) likely reflecting allelic polymorphic conditions in the heterokaryotic (n+n) basidiomata. Deletions or insertions are mentioned in square brackets. The locality/position of the markers are numbered with respect to their position either in the 5'-3' ITS1–5.8S–ITS2 sequence data (starting from tygaatt) of the taxon to be characterized or in the final alignment deposited in TreeBASE for the clades or higher taxa characterization.

## Results

### Phylogenetic analyses

The phylogenetic analyses comprised of 42 sequences that were newly generated (30 ITS, 06 LSU, and 06 TEF1) from 30 specimens. These 30 specimens belonged to six species including five new species described in this study and a new record (*A. robustulus*) from Pakistan. ITS sequences of all the specimens were included for the first time in phylogenetic analyses. The alignment for ITS dataset consisted of total sequence of 158 specimens and 718 characters (Fig 1). Among these, seven samples were used as outgroup belonging to seven different species of: *A*. sect. *Leucocarpi* (*A. leucocarpus*), *A*. subg. *Minoriopsis* (*A. rufoaurantiacus* and *A*. sp. ZRLWXH3064), *A*. sect. *Pantropicales* (*A. candidolutescens*, *A*. sp. LAPAM14 and *A*. sp.

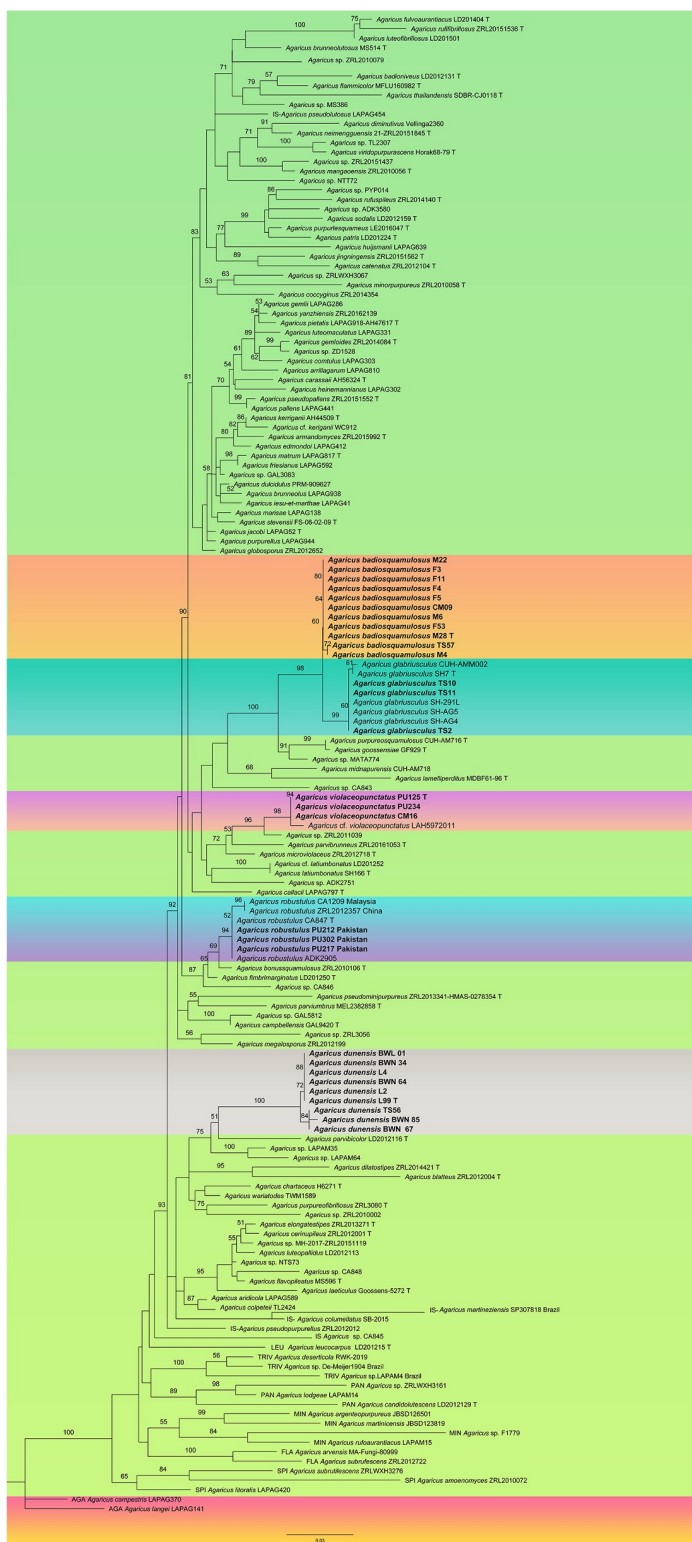

**Fig 1. Maximum likelihood tree of species belonging to *A.* sect. *Minores* generated from ITS sequence data.**
*Agaricus campestris* is used as outgroup. Bootstrap support values are indicated (BS). New species and new record from Pakistan are in bold. T = type specimen.

ZRLWXH3161) and *A.* sect. *Agaricus* (*A. campestris*). All samples with their GenBank accession numbers are listed in Table 1 and references in Table 2.

The alignment for multi-gene analyses consisted of sequences (ITS/LSU/TEF1) of 105 sequences and 2052 characters (Fig 2). The five samples used as outgroup belonging to five different species of: *A.* sect. *Leucocarpi* (*A. leucocarpus*), *A.* sect. *Pantropicales* (*A. candidolutescens*, *A.* sp. LAPAM14 and *A.* sp. ZRLWXH3161) and *A.* sect. *Agaricus* (*A. campestris*). *Agaricus huijsmanii*, a European species, is separated from all the clades in multigene phylogenetic analyses as referenced in He et al. 2017 [32] also, however, this taxon got aligned within *A.* sect. *Minores* in ITS phylogenetic analyses. The remaining 99 samples belonged to 96 species or putative species of *A.* sect. *Minores* (Table 1).

For both multigene and ITS analyses, the topologies of the phylogenetic trees using the Maximum likelihood (ML) or the Bayesian inference (BI) methods were highly similar. The bootstrap support values (BS) and Bayesian posterior probabilities (PP) are reported in the ML phylogenetic trees of Fig 1 for ITS analyses, and Fig 2 for multigene analyses. The multigene data of all the newly reported species and those available of previously reported samples were completed in our multigene analyses, however, still there remain many species that only have ITS sequence data. For this reason, we presented first the phylogeny of the sections based on multigene-analyses. The ITS analyses remain useful mainly at the species level to delimit new taxa and identify their most closely related taxa.

## Taxonomy

*Agaricus badiosquamulosus* **H. Bashir & Khalid sp. nov**. (Figs 3 and 4).

**Species–specific ITS markers**: aatcttCttcccg@200, attccTTggagca@420–421

**Etymology:** The specific epithet '*badio*' means in Latin brown and '*squamulosus*' means scales, '*badiosquamulosus*' referred to the brown colored squamules on the pileus.

**Holotype:** PAKISTAN, Punjab, Lahore, solitary on rich loamy soil along Pak. Motorway (M3) near *Eucalyptus camadulensis*, 217 m a.s.l., 01 September 2016, Hira Bashir, **M28** (LAH35751), (GenBank # ON137221(nrITS).

**Diagnosis:** This species is characterized by having a dark brown pileus covering with purplish tinged squamules, immediately gives strong fulvescent discoloration when bruised. Annulus membranous, single edged with erroded edges in most of specimens and superous. Basidia frequently bisporic to tetrasporic and monosporic observed in only one collection, cheilocystidia abundant, multi-septate and broadly clavate.

**Description: Pileus** 3–5 cm in diam., parabolic at first then convex, slightly umbonate, dark brown (4.3R 2.3/2.4) at disc, dark brown with purple tinged (4.5RP 6.6/3.4) slightly appressed squamules when young (M28), disc completely covered with dark brown (0.5YR 4/ 2.8) squamules in younger sporocarp (F3), later scattered towards margin on creamy white (1.3PB 9.6/2.2) background outside disc or umbo. Surface fresh and fragile, light yellow surface observed in F3 and F4 after rain. **Margins** appendiculate, slightly wavy at maturity in some specimens (M28), vaguely exceeding the lamellae, ruptured at maturity, orange-yellow (10YR 8.3/2.5) discoloration when rubbed. **Lamellae** light pink (2.8R 8.1/3.6) to mouse-grey (9RP 6.3/0.8), then light (8.6R 6.5/3.2) to dark brown (3.3YR 3.5/1.8), free, crowded, with intercalated lamellulae in 2–3 tiers, sometimes erroded. **Stipe** 3.0–6 × 0.5–1 cm, cylindrical in some specimens, mostly curved at maturity, slightly bulbous at base, partially stuffed, provided with annulus in its upper part near lamellae, white (N10) with concolorous longitudinal striations above and below the annulus, immediately discoloring yellow-brown when rubbed, ends with rhizomorphs. **Annulus** superous, single-edged, membranous, fragile, non-persistent in some specimens (M22, M28, F53, CM09) only ring zone observed, or pendent (F4, F5, M4), white

**Table 1. All the representatives of *Agaricus* sect. *Minores* sequences of known and unnamed species are mentioned along with the outgroup and other sections samples.**

| Taxon | | | Reference | | ITS | LSU | TEF1 |
|---|---|---|---|---|---|---|---|
| **13 [4-3-6] *Agaricus* subg. *Minoriopsis* Linda J. Chen, L.A. Parra, Callac, Angelini & Raspé** | 2017 | | 49 | | | | |
| **2 [1-1-0] *Agaricus* sect. *Minoriopsis* (Linda J. Chen, L.A. Parra, Callac, Angelini & Raspé) L.A. Parra, Angelini, Linda J. Chen & Callac** | 2018 | | 60 | | | | |
| *Agaricus argenteopurpureus* L.A. Parra, Angelini & Callac | 2018 | CA | 60 | LAPAM28 (Dominican Rep.) T | KX671700 | KX671710 | KX671707 |
| *Agaricus martinicensis* Pegler | (1983) | CA | 60 | JBSD123819 (Dominican Rep.) | KX671699 | KX671709 | KX671706 |
| **11 [3-2-6] *Agaricus* sect. *Kerrigania* L.A. Parra, Angelini, B. Ortiz, Linda J. Chen & Callac** | 2018 | | 60 | | | | |
| *Agaricus globocystidiatus* Drewinski & M.A. Neves | 2017 | SA NA | 50 | MPD29 (Brazil) T | MF188247 | | |
| *Agaricus guzmanii* Linda J. Chen & G. Mata | 2019 | NA | 64 | LD201811 (Mexico) T | MK467614 | | |
| *Agaricus porphyropos* L.A. Parra, Angelini & B. Ortiz | 2018 | CA | 60 | JBSD126494 (Dominican Rep.) T | KX671703 | | |
| *Agaricus floridanus* Peck | (1911) | NA | C 60 | JH-1 (USA, NY) | KM349609 | | |
| *Agaricus rufoaurantiacus* Heinem. | (1961) | CA | 60 | LAPAM15 (Dominican Rep.) | KT951313 | KX671708 | KT951641 |
| *Agaricus* sp. | | NA | 60 | RMC-1256 (USA) | KM349611 | | |
| *Agaricus* sp. | | SA | 60 | LAPAM66 (Brazil) | KX671702 | | |
| *Agaricus* sp. | | CA | C 60 | F1779 (Martinique, France) | JF727853 | | |
| *Agaricus* sp. | | SA | C 60 | CJL030902.05 (Guiana, France) | JF727869 | | |
| *Agaricus* aff. *rufoaurantiacus* | | CA | C 60 | CL/GUAD05.099 (Guadeloupe, France) | JF727857 | | |
| *Agaricus* aff. *rufoaurantiacus* | | SA | C 60 | LAPAM1 (Venezuela) | JF797183 | | |
| **122 [72-10-40] *Agaricus* subg. *Minores* (Fr.) R.L. Zhao & Moncalvo** | 2016 | | 47 | | | | |
| **3 [1-0-2] *Agaricus* sect. *Leucocarpi* Linda J. Chen & Callac** | 2017 | | 49 | | | | |
| *Agaricus leucocarpus* Linda J. Chen, Callac, R.L. Zhao & K.D. Hyde | 2017 | AS | 49 | LD201215 (Thailand) T | KU975101 | KX083981 | KX198048 |
| *Agaricus* sp. | | AS | 51 | ZRLWXH3064 (China, Jian) | KX657010 | | |
| *Agaricus* sp. | | AS | 51 | ZRLWXH3316 (China) | KX657015 | | |
| **4 [2-0-2] *Agaricus* sect. *Pantropicales* L.A. Parra, Angelini, B. Ortiz, Linda J. Chen & Callac** | 2018 | | 60 | | | | |
| *Agaricus candidolutescens* Linda J. Chen & R.L. Zhao | 2016 | AS | 47 | LD2012129 (Thailand) T | KT951391 | KT951525 | KT951616 |
| *Agaricus lodgeae* L.A. Parra, Angelini & B. Ortiz | 2018 | CA | 50 60 | LAPAM14 (Dominican Rep.) | KT951312 | | KT951613 |
| *Agaricus* sp. | | CA | 60 | LAPAM68 (Dominican Rep.) | MF511150 | | |
| *Agaricus* sp. | | AS | 60 | ZRLWXH3161 (China, Gua) | KT951391 | KT951526 | KT951615 |
| **115 [69-10-36] *Agaricus* sect. *Minores* (Fr.) Henn.** | (1898) | | | | | | |
| **28 [18-6-4] Clade I** | | | | | | | |
| *Agaricus armandomyces* M.Q. He & R.L. Zhao | 2017 | AS | 51 | ZRL2015992 (China, Yun) T | KX684860 | KX684882 | KX684906 |
| *Agaricus arrillagarum* L.A. Parra, S. Serrano & Geml | 2013 | EU | 28 | LAPAG810 (Spain) | KF447900 | KX083985 | KT951592 |

*(Continued)*

**Table 1.** (Continued)

| Taxon | | | Reference | | ITS | LSU | TEF1 |
|---|---|---|---|---|---|---|---|
| *Agaricus carassaii* Faraoni, L.A. Parra & Suriano | 2021 | EU | 76 | AH56324 (Italy) T | AH56324 | | |
| *Agaricus edmondoi* L.A. Parra, Cappelli & Callac | 2013 | EU | 28 | LAPAG412 (Spain) | KT951326 | KT951481 | KT951590 |
| *Agaricus friesianus* L.A. Parra, Olariaga & Callac | 2013 | AS EU | 28 | LAPAG592 (France) | KT951316 | KX083992 | KT951594 |
| *Agaricus gemlii* L.A. Parra, Arrillaga, M.Á. Ribes & Callac | 2013 | AS EU | 28 | LAPAG286 (Spain) | KU975079 | KX083988 | KX198055 |
| *Agaricus gemloides* M.Q. He & R.L. Zhao | 2015 | AS | 36 | ZRL2014084 (China, Yun) T | KT633271 | KX641405 | KX684986 |
| *Agaricus globosporus* M.Q. He & R.L. Zhao | 2017 | AS | 51 | ZRL2012652 (China, Tib) | KX657036 | KX656976 | KX684967 |
| *Agaricus iesu-et-marthae* L.A. Parra (≡ *Agaricus lutosus* var. *macrosporus* L.A. Parra) | 2013 | EU | 28 | LAPAG41 (Spain) | KF447904 | | |
| *Agaricus jacobi* L.A. Parra, A. Caball. & Callac | 2013 | EU | 28 | AH44505 (Spain) T | KF447895 | KX083996 | KX198061 |
| *Agaricus kerriganii* L.A. Parra, B. Rodr., A. Caball., M. Martín-Calvo & Callac | 2013 | EU | 28 | AH44509 (Spain) T | KF447893 | KX083999 | KX198066 |
| *Agaricus marisae* L.A. Parra & Callac | 2013 | EU | 28 | LAPAG138 (Spain) | KU975083 | KX083998 | KX198065 |
| *Agaricus matrum* L.A. Parra, A. Caball., S. Serrano, E. Fern. & Callac | 2013 | EU | 28 | AH44506 (Spain) T | KF447896 | KX083991 | KX198058 |
| *Agaricus pallens* (J.E. Lange) L.A. Parra (≡ *Psalliota rubella* f. *pallens* J.E. Lange) | 2013 | EU | 28 | LAPAG441 (Spain) | KF447898 | | KX198067 |
| *Agaricus pietatis* L.A. Parra & A. Caball. | 2017 | EU | 54 | LAPAG918 (Spain) T | MF568546 | | |
| *Agaricus pseudopallens* M.Q. He & R.L. Zhao | 2017 | AS | 51 | ZRL20151552 (China, Zhe) T | KX684874 | KX684891 | |
| *Agaricus stevensii* Kerrigan | 2016 | NA | 44 | FS 06-02-09 (USA, Ca) T | KJ877785 | | |
| *Agaricus yanzhiensis* M.Q. He, K.D. Hyde & R.L. Zhao | 2018 | AS | 58 | ZRL20162060 (China, Gan) | MG137002 | MG196348 | MG196352 |
| *A. dulcidulus* Schulzer | (1874) | EU | 51 | PRM909627 (Czech Republic) | KF447894 | | KX198064 |
| *A. purpurellus* F.H. Møller | (1952) | AS EU | 51 | LAPAG944 (Czech Republic) | KU975076 | KX083994 | KX198060 |
| *A. brunneolus* (J.E. Lange) Pilát | (1951) | EU | 51 | LAPAG938 (Spain) | KU975082 | KX083997 | KX198062 |
| *A. comtulus* Fr. | (1838) | EU | 47 | LAPAG724 (Spain) | KT951332 | KT951448 | KT951593 |
| *A. heinemannianus* Esteve-Rav. | (1999) | EU | 51 | LAPAG302 (Spain) | KF447906 | | KX198056 |
| *A. luteomaculatus* F.H. Møller | (1952) | EU | 51 | LAPAG331 (France) | KF447901 | | KX198053 |
| *Agaricus* cf. *kerriganii* L.A. Parra, B. Rodr., A. Caball., M. Martín-Calvo & Callac | (2013) | NA | 51 | WC912 (USA Ca) | AY484681 | | |
| *Agaricus* sp. | | NA | 51 | GAL3083 (USA, Ala) | EF460374 | EF460399 | |
| *Agaricus* sp. | | AS | 51 | ZD1528 (China, Yun) | KU975104 | KX083987 | KX198054 |
| *Agaricus* sp. | | AS | 51 | ZRL2014380 (China, Yun) | KX656998 | KX656932 | KX685000 |
| 4 [2-0-2] Clade II | | | | | | | |
| *Agaricus parvibicolor* Linda J. Chen, R.L. Zhao & K.D. Hyde | 2015 | AS | 34 | LD2012116 (Thailand) T | KP715162 | KX084016 | KX198075 |
| *Agaricus dunensis* | | | | L99 T (Pakistan) | **ON137217** | **OP835847** | **OP903344** |
| *Agaricus dunensis* | | | | L2 (Pakistan) | **ON137218** | - - | |
| *Agaricus dunensis* | | | | LS4 (Pakistan) | **ON137219** | - - | |
| *Agaricus dunensis* | | | | TS56 (Pakistan) | **ON137220** | - - | |
| *Agaricus dunensis* | | | | BWL-01 (Pakistan) | **ON158598** | - - | |
| *Agaricus dunensis* | | | | BWN-34 (Pakistan) | **ON158597** | - - | |
| *Agaricus dunensis* | | | | BWN-64 (Pakistan) | **ON158596** | **OP835848** | **OP903345** |

(*Continued*)

**Table 1.** (Continued)

| Taxon | | | Reference | | ITS | LSU | TEF1 |
|---|---|---|---|---|---|---|---|
| *Agaricus dunensis* | | | | BWN-67 (Pakistan) | **ON158600** | - - | - - |
| *Agaricus dunensis* | | | | BWN-85 (Pakistan) | **ON158599** | - - | - - |
| *A.* sp. | | | | LAPAM35 (Dominican Rep) | MF511126 | | |
| *A.* sp. | | | | LAPAM64 (Dominican Rep) | MF511148 | | |
| 10 [7-0-3] Clade III | | | | | | | |
| *Agaricus badioniveus* Linda J. Chen, R.L. Zhao & K.D. Hyde | 2017 | AS | 49 | LD2012131 (Thailand) T | KU975117 | | KX198072 |
| *Agaricus brunneolutosus* Linda J. Chen, Kanun. & K.D. Hyde | 2017 | AS | 49 | MS514 (China Yun) T | KU975111 | KX084006 | |
| *Agaricus flammicolor* Linda J. Chen, Callac, R.L. Zhao & K.D. Hyde | 2017 | AS | 49 | LD201502 (Thailand) T | KU975114 | KX084009 | KX198042 |
| *Agaricus fulvoaurantiacus* Linda J. Chen & Kanun. | 2017 | AS | 49 | LD201404 (China Yun)) T | KU975107 | KX084002 | KX198069 |
| *Agaricus luteofibrillosus* M.Q. He, Linda J. Chen & R.L. Zhao | 2016 | AS | 45 | LD201501 (Thailand) | KU975108 | KX084003 | KX198041 |
| *Agaricus rufifibrillosus* M.Q. He & R.L. Zhao *(rufusfibrillosus)* | 2017 | AS | 52 (58) | ZRL 20151536 (China Zhe) T | KX684878 | | |
| *Agaricus thailandensis* C. Jaichliaw & S. Lumyong (**needing validation**) | 2021 | AS | 74 | SDBR-CJ0118 (Thailand) T | MW255675 | MW255677 | MW264832 |
| *Agaricus* sp. | | AS | 58 | ZRL2010079 (China, Yun) | KX657046 | KX656951 | KX684950 |
| *Agaricus* sp. | | AS | 58 | CA935 (Thailand) | KU975085 | KX084036 | KX198034 |
| *Agaricus* sp. | | AS | 58 | MS386 (China, Yun) | KU975113 | KX084008 | KX198044 |
| 6 [4-0-2] Clade IV | | | | | | | |
| *Agaricus patris* Linda J. Chen, Callac, K.D. Hyde & R.L. Zhao | 2017 | AS | 49 | LD201224 (Thailand) T | KU975118 | KX084012 | KX198073 |
| *Agaricus purpureosquameus* M.Q. He & R.L. Zhao *(purpurlesquameus)* | 2017 | AS | 52 (58) | LE2016047 T | MF611640 | | |
| *Agaricus rufipileus* M.Q. He & R.L. Zhao *(rufuspileus)* | 2017 | AS | 51 (58) | ZRL2014140 (China, Yun) T | KX656991 | KX656937 | KX684991 |
| *Agaricus sodalis* Linda J. Chen, R.L. Zhao & K.D. Hyde | 2015 | AS | 34 | LD2012159 (Thailand) T | KP715161 | KX084014 | KX198074 |
| *Agaricus* sp. | | AS | 51 | PYP014 (Thailand) | KU975091 | | |
| *Agaricus* sp. | | AF | 51 | ADK3580 (Bénin) | KU975097 | | |
| 2 [1-0-1] Clade V | | | | | | | |
| *Agaricus megalosporus* Linda J. Chen, R.L. Zhao, Karun. & K.D. Hyde | 2012 | AS | 23 | ZRL2012199 (China Yun) | KT951367 | KT951470 | KT951595 |
| *Agaricus* sp. | | AS | 51 | ZRL3056 (Thailand) | JF691541 | KX084020 | |
| 7 [4-0-3] Clade VI | | | | | | | |
| *Agaricus microviolaceus* M.Q. He & R.L. Zhao | 2017 | AS | 51 | ZRL2012718 (China, Yun) T | KX657033 | KX656980 | KX684971 |
| *Agaricus latiumbonatus* S. Hussain | 2019 | AS | 65 | SWAT SH166 (Pakistan) T | MK751861 | MK751858 | |
| *Agaricus parvibrunneus* M.Q. He, K.D. Hyde & R.L. Zhao | 2018 | AS | 58 | ZRL20161053 (China, Beijing) T | MG137001 | MG196345 | MG196351 |
| *Agaricus violaceopunctatus* | | | | PU125 T (Pakistan) | ON158593 | OP835850 | - - |
| *Agaricus violaceopunctatus* | | | | PU234 (Pakistan) | ON158594 | - - | - - |
| *Agaricus violaceopunctatus* | | | | CM16 (Pakistan) | ON158595 | - - | OP903347 |
| *Agaricus* sp. | | | | LAH5972011 | KU170540 | | |
| *Agaricus* sp. | | | 51 | ZRL2011039 (China, Yun) | KT951351 | KT951449 | KT951606 |
| *Agaricus* sp. | | | 49 | ADK2751 (Bénin) | JF514519 | | |

*(Continued)*

**Table 1.** (Continued)

| Taxon | | | Reference | | ITS | LSU | TEF1 |
|---|---|---|---|---|---|---|---|
| 4 [1-2-1] Clades VII + VIII | | | | | | | |
| *Agaricus neimengguensis* M.Q. He & R.L. Zhao | 2017 | AS | 51 | ZRL20151845 T | KX684870 | KX684902 | KX684924 |
| *Agaricus diminutivus* Peck | (1874) | AN | 44 51 | Vellinga2360 (USA, Wash) | AF482831 | AF482877 | |
| *Agaricus viridopurpurascens* Heinem. | (1974) | OC | 51 | Horak68/79 (New Zeland) T | JF514525 | | |
| *Agaricus* sp. | | OC | 51 | TL2307 (Australia) | JF495058 | | |
| 6 [5-0-1] Clade IX | | | | | | | |
| *Agaricus blatteus* M.Q. He & R.L. Zhao | 2017 | AS | 51 | ZRL2012004 (China, Yun) T | KT951355 | KT951457 | KT951608 |
| *Agaricus dilatostipes* M.Q. He & R.L. Zhao | 2017 | AS | 51 | ZRL2014450 (China, Yun) P | KX656999 | KX656941 | KX685003 |
| *Agaricus catenatus* M.Q. He & R.L. Zhao | 2017 | AS | 51 | ZRL2012104 (China, Yun) T | KX657023 | KX656963 | KX684957 |
| *Agaricus jingningensis* M.Q. He & R.L. Zhao | 2017 | AS | 51 | ZRL20151562 (China, Zhe) T | KX684877 | KX684895 | KX684917 |
| *Agaricus mangaoensis* M.Q. He & R.L. Zhao. | 2017 | AS | 51 | ZRL2010056 (China, Yun) T | KX657042 | KX656956 | KX684946 |
| *Agaricus* sp. | | AS | 51 | ZRL20151437 (China, Zhe) | KX684876 | KX684892 | KX684914 |
| 9 [7-0-2] Clades X + XIII | | | | | | | |
| *Agaricus chartaceus* T. Lebel | 2012 | OC | 25 | H6271 (Australia) T | JF495048 | | |
| *Agaricus lamelliperditus* T. Lebel & M.D. Barrett | 2013 | OC | 24 | MDBF61/96 (Australia) T | JX984559 | | |
| *Agaricus midnapurensis* Tarafder, A.K. Dutta & K. Acharya | 2022 | AS | 79 | CUH AM718 (India) T | OL467539 | | |
| *Agaricus coccyginus* M.Q. He & R.L. Zhao | 2016 | AS | 45 | ZRL2014354 (China, Yun) | KU245981 | KX656936 | KX684998 |
| *Agaricus minipurpureus* M.Q. He & R.L. Zhao *(minorpurpureus)* | 2017 | AS | 51 (58) | ZRL2010058 (China, Yun) T | KX657043 | KX656953 | KX684947 |
| *Agaricus* sp. | | | 51 | ZRLWXH3067 (China, Jia) | KT951387 | KT951497 | KT951611 |
| *Agaricus purpureofibrillosus* Linda J. Chen, R.L. Zhao & K.D. Hyde | 2017 | AS | 49 | ZRL3080 (Thailand) T | JF691542 | KX084021 | |
| *Agaricus wariatodes* (Grgur.) T. Lebel (≡ *Endoptychum wariatodes* Grgur.) | 2012 | OC | 25 | TWM1589 (Australia) | JF495052 | JF495030 | |
| *Agaricus* sp. | | AS | 51 | ZRL2010002 (Thailand) | KX657041 | KX656954 | KY427449 |
| 10 [5-0-5] Clade XI | | | | | | | |
| *Agaricus cerinupileus* M.Q. He & R.L. Zhao *(cerinupileus)* | 2017 | AS | 51 (58) | ZRL2012001 (China, Yun) T | KX657021 | KX656957 | KX684953 |
| *Agaricus elongatestipes* M.Q. He & R.L. Zhao | 2017 | AS | 51 | ZRL2013271 (China, Yun) T | KX657002 | KX656946 | KX684975 |
| *Agaricus flavopileatus* Linda J. Chen, Kanun. & Callac | 2017 | AS | 49 | MS596 (China, Yun) T | KU975121 | KX084022 | KX198078 |
| *Agaricus laeticulus* Callac, L.A. Parra, L.J. Chen & Raspé (≡ *Agaricus laeticolor* Heinem. & Gooss.-Font.) | 2017 (1956) | AF | 49 | Goossens5272 (DR Congo) T | KX671705 | | |
| *Agaricus luteopallidus* Linda J. Chen, Kanun., R.L. Zhao & K.D. Hyde | 2017 | AS | 49 | LD2012113 (Thailand) | KU975124 | KX084026 | KX198080 |
| *Agaricus* sp. | | AS | 49 | CA848 (Thailand) | JF727864 | KT951445 | KT951605 |
| *Agaricus* sp. | | AS | 49 | NTS73 (Thailand) | KU975099 | | |
| *Agaricus* sp. | | AS | 51 | ZRL20151119 (China, Sich) | KX684855 | KX684890 | KX684913 |
| *Agaricus* sp. | | AS | 49 | ZRLLD013 (Thailand) | KT951384 | KT951516 | KT951604 |

(*Continued*)

**Table 1.** (Continued)

| Taxon | | | Reference | | ITS | LSU | TEF1 |
|---|---|---|---|---|---|---|---|
| *Agaricus* sp. | | AS | 49 | ZRL2011156 (China, Yun) | KT951352 | KT951480 | KT951603 |
| 2 [2-0-0] Clade XII | | | | | | | |
| *Agaricus aridicola* Geml, Geiser & Royse ex Mateos, J. Morales, J. A. Muñoz, Rey & C. Tovar (≡ Montagnites dunalii Fr.) | 2004 | EU | 10 | LAPAG589 (Spain) | KT951331 | KX084027 | KX198081 |
| *Agaricus colpeteii* T. Lebel | 2013 | OC | 24 | TL2424 (Australia) T | JX984565 | | |
| 5 [3-0-2] Clade XIV | | | | | | | |
| *Agaricus bonisquamulosus* M.Q. He & R.L. Zhao (*bonussquamulosus*) | 2017 | AS | 51 (58) | ZRL2010106 (China, Yun) T | KX657047 | KX656950 | KX684951 |
| *Agaricus robustulus* Linda J. Chen, Callac, L.A. Parra, K.D. Hyde & De Kesel | 2017 | AF AS | 49 | CA847 '(Thailand) T | KU975086 | KX084034 KX198039 | KX198039 |
| *Agaricus robustulus* | | | | PU217 (Pakistan) | ON133841 | - - | - - |
| *Agaricus robustulus* | | | | PU212 (Pakistan) | ON133842 | - - | - - |
| *Agaricus robustulus* | | | | PU302 (Pakistan) | ON133843 | - - | - - |
| idem A. sp. | | AF | 49 | ADK2905 (Bénin) | JF514520 | | |
| idem A. sp. | | AS | 49 | ZRL2012357 (China) | KT951369 | KT951496 | KT951610 |
| idem | | AS | 49 | AK075 (Malaysia) | KU975088 | | |
| idem | | | | Pakistan | | | |
| *Agaricus fimbrimarginatus* Linda J. Chen, Callac & K.D. Hyde | 2017 | AS | 49 | LD201250 (Thailand) T | KU975119 | KX084017 | KX198076 |
| *Agaricus* sp. | | AS | C | CA846 (Thailand) | JF727865 | | |
| *Agaricus* sp. | | AS | C | ZRL2044 (Thailand) | JF691540 | | |
| 5 [3-0-2] Clade XV | | | | | | | |
| *Agaricus parviumbrus* Broadbridge, Boxshall & T. Lebel, sp. nov. | 2022 | OC | 81 | MEL:2382858 (Australia) T | KP012732 | | |
| *Agaricus campbellensis* Geml, Laursen & D.Lee Taylor | 2007 | OC | 14 | GAL9420 (New Zeland) T | DQ232644 | DQ232657 | |
| *Agaricus pseudominipurpureus* M.Q. He, K.D. Hyde & R.L. Zhao | 2018 | AS | 58 | ZRL2013341 (China, Yun) T | MG137000 | MG196343 | MG196350 |
| *Agaricus* sp. | | NA | 51 | GAL5812 (USA, Alaska) | EF460364 | EF460389 | |
| *Agaricus* sp. | | OC | 51 | PDD68575 (New Zeland) | AF059224 | AF059224 | |
| 5 [3-1-1] New clade "goossensiae" | | | | | | | |
| *Agaricus glabriusculus* S. Hussain | 2019 | AS | 65 | SH7 (Pakistan) T | MK751852 | MK751811 | |
| *Agaricus glabriusculus* | | | | TS10 | ON158590 | OP835849 | OP903346 |
| *Agaricus glabriusculus* | | | | TS2 | ON158591 | - - | - - |
| *Agaricus glabriusculus* | | | | TS11 | ON158592 | - - | - - |
| *Agaricus purpureosquamulosus* Tarafder, A.K. Dutta & K. Acharya | 2022 | AS | 79 | CUH AM716 (India) T | OL467541 | | |
| *Agaricus badiosquamulosus* | | | | F3 (Pakistan) | ON137225 | - - | - - |
| *Agaricus badiosquamulosus* | | | | F4 (Pakistan) | ON137226 | - - | - - |
| *Agaricus badiosquamulosus* | | | | F5 (Pakistan) | ON137227 | - - | - - |
| *Agaricus badiosquamulosus* | | | | F11 (Pakistan) | ON137228 | - - | - - |
| *Agaricus badiosquamulosus* | | | | F53 (Pakistan) | ON137229 | - - | - - |
| *Agaricus badiosquamulosus* | | | | M4 (Pakistan) | ON137222 | - - | - - |
| *Agaricus badiosquamulosus* | | | | M6 (Pakistan) | ON137223 | - - | - - |
| *Agaricus badiosquamulosus* | | | | M22 (Pakistan) | ON137224 | - - | - - |
| *Agaricus badiosquamulosus* | | | | M28 T (Pakistan) | ON137221 | OP831149 | OP903342 |
| *Agaricus badiosquamulosus* | | | | CM09 (Pakistan) | ON137231 | OP831150 | OP903343 |
| *Agaricus badiosquamulosus* | | | | TS57 (Pakistan) | ON137230 | - - | |

(*Continued*)

**Table 1.** (Continued)

| Taxon | | | Reference | | ITS | LSU | TEF1 |
|---|---|---|---|---|---|---|---|
| *Agaricus goossensiae* Heinem. | (1956) | | | Goossens-Fontana 929 (RD Congo) T | | | |
| *Agaricus* sp. | | | C | MATA774 '(Mexico) | JF727871 | | |
| 12 [4-1-7] Unclassified | | | | | | | |
| *Agaricus callacii* L.A. Parra, Iglesias, Fern. Vincente & Oyarzabal | 2013 | EU | 28 | AH42929 (Canary Island Spain) T | KF447899 | KX083984 | KX198051 |
| *Agaricus columellatus* (Long) R.M. Chapm., V.S. Evenson & S.T. Bates (≡ *Araneosa columellata* Long) | 2016 | NA | 40 | SB-2015 (USA, Ari) | KJ912899 | KJ912900 | |
| *Agaricus huijsmanii* Courtec. (≡ *Agaricus niveolutescens* Huijsman) | 2008 | EU | 16 | LAPAG639 (spain) | KF447889 | KT951444 | KT951571 |
| *Agaricus pseudopurpurellus* M.Q. He & R.L. Zhao | 2017 | AS | 51 | ZRL2014063 (China, Yun) T | KX656988 | KX641404 | KX684985 |
| *Agaricus pseudolutosus* (G. Moreno, Esteve-Rav., Illana & Heykoop) G. Moreno, L.A. Parra, Esteve-Rav. & Heykoop | (1999) | EU | 51 | LAPAG454 (Spain) | KT951329 | KT951453 | KT951602 |
| *Agaricus* sp. | | AS | 51 | NTT72 (Thailand) | JF514539 | | |
| *Agaricus* sp. | | AS | 51 | CA845 (Thailand) | KU975084 | KX084033 | KX198035 |
| *Agaricus* sp. | | AS | 51 | LD201252 (Thailand) | KU975103 | KX083983 | KX198050 |
| *Agaricus* sp. | | AS | 51 | CA843 (Thailand) | JF727866 | KX084029 | KX198040 |

Table 1. All the representatives of Agaricus sect. Minores sequences of known and unnamed species are mentioned along with the outgroup and other sections samples.

(A) the newly named species since 2000 have their names ajusted to the left in the column (names in black letters for new species and green letters for renamed species)

(B) the newly named species since 2000 have their names ajusted to the left in the column (names in black letters for new species with no colored background and with green background for renamed species)

(C) species named before 2000 are adjusted to the right of the column and are in red background

(D) the unnamed or provisionally named species (cf. or aff.) are on the left and in blue background

---for samples that are not the type but which have more sequence data (some are paratype P but we have not verified all)

---for samples of the same species later reported from distant countries (generally different continents indicted with initial AF, AS, EU, NA, SA, OC, and CA for the Caribbean)

(N10), fibrillose, edges uneven. **Context** 2–4 mm wide, thick, white, immediately becomes strongly fulvescent (7.7YR 4.3/6.5) when bruised. **Odor** almond-like, strong. (Fig 3).

**Macrochemical Reactions:** KOH reaction positive, strong yellow. Schäffer's reaction positive, strong reddish-orange.

**Basidiospores** (5.5–) 6–6.5 (–7) × (4.2–) 4.5–4.8 (–5.1) μm, [avX = $6.2 \pm 0.67 \times 4.6 \pm 0.45$ μm, $Q_m$ = 1.32, n = 20 × 10], subglobose to broadly ellipsoid, light to dark brown in KOH, smooth with a prominent apiculus, without apical pore and with granular content. **Basidia** 16–22 × 7–8.5 μm, narrowly clavate in most specimens, broadly clavate in F53 only, with abundant olivaceous granular content, frequently bisporic to tetrasporic, monosporic observed in F11. **Cheilocystidia** abundant, short and multiseptate and broadly clavate to rarely simple with terminal elements 11.5–20.5 × 6.5–11 μm, [avX = $15.9 \pm 4.6 \times 8.8 \pm 2.3$ μm], with ante-terminal elements mostly short, broader, cylindrical, 4–11 × 2.5–6 μm, olivaceous granular content observed but not abundant. **Pleurocystidia** absent. **Pileipellis** constituted by interwoven hyphae 4–10 μm in diam., frequently septate, branched, slightly constricted at septa, smooth, hyaline in KOH, terminal elements with rounded comparatively narrow ends. **Stipitipellis** constituted by hyphae 3–6.5 μm in diam., cylindrical, septate, unbranched, hyaline in KOH, no constriction at septa, terminal elements with narrow rounded tips. (Fig 4).

**Table 2.** References.

| | | |
|---|---|---|
| 2000 | 1 | Albertó E, Sannazzaro A, Moreno G (2000) *Agaricus heinemannii* a new species from Argentina. Micol. Veg. Medit. 15:71–78 |
| 2001 | 2 | Blanco-Dios JB (2001) Agaricales des dunes de Galice (Nord-Ouest de l'Espagne) 1: Agaricus freirei, sp. nov. Doc. Mycol. 31(121):27–34. |
| 2002 | 3 | Lanconelli L (2002) Agaricus padanus sp. nov. Rivista di Micologia. 1:29–37. |
| 2002 | 4 | Parra LA, Arrillaga P (2002) Agaricus laskibarii. A new species from French coastal sand-dunes of Seignosse. Doc. Mycol. 31(124):33–38. |
| 2002 | 5 | Wasser SP, Didukh MY, de Amazonas MAL, Nevo E, Stamets P, da Eira AF. (2002). Is a widely cultivated culinary-medicinal Royal Sun Agaricus (the Himematsutake Mushroom) indeed Agaricus blazei Murrill? International Journal of Medicinal Mushroom 4:267–290. |
| 2003 | 6 | Callac P., Jacobe de Haut I., Imbernon M., Guinberteau J. et Theochari I. (2003). A novel homothallic variety of Agaricus bisporus comprises rare tetrasporic isolates from Europe, Mycologia 95: 222–231 |
| 2003 | 7 | Vellinga EC, Kok RPJ de, Bruns TD (2003) Phylogeny and Taxonomy of Macrolepiota (Agaricaceae). Mycologia 95:442–456. |
| 2004 | 8 | Callac P, Mata G (2004) Agaricus tollocanensis, une nouvelle espèce de la section Xanthodermatei trouvée au Mexique. Doc. Mycol. 33(132):31–35. |
| 2004 | 9 | Lanconelli L., Nauta M.M. (2004) Un nuovo taxon dall'Italia: *Agaricus rufotegulis* var. *hadriaticu*s. Bolletinno del Gruppo Micolgica G. Bresadola, Nuova Serie 47 (2): 15–22. |
| 2004 | 10 | Mateos Izquierdo A, Morales Pulido J, Muñoz Mohedano J, Rey Expósito R., Carlos Tovar Breña C (2009) Agaricomycetes gasteroides de interés en Extremadura. Boletín Informativo de la Sociedad Micológica Extremeña 9(20): 41–53 |
| 2005 | 11 | Callac P, Guinberteau J (2005) Morphological and molecular characterization of two novel species of Agaricus section Xanthodermatei. Mycologia 97:416–424. |
| 2005 | 12 | Parra LA (2005) Nomenclatural study of the genus Agaricus L. (Agaricales, Basidiomycotina) of the Iberian Peninsula and Balearic Islands. Cuadernos de Trabajo de Flora Micológica Ibérica (Madrid) 21:3–101. |
| 2006 | 13 | Zuccherelli A. (2006) I Funghi della pinete delle zone mediterranee. Longo Angelo Editore, Ravenna. |
| 2007 | 14 | Geml J, Laursen GA, Nusbaum HC, Taylor DL (2007) Two new species of Agaricus from the Subantarctic. Mycotaxon 100:193–208. |
| 2007 | 15 | Ludwig E. (2007) Pilzkompendium 2. Fungicon-Verlog. Berlin. |
| 2008 | 16 | Courtecuisse R. (2008) Nouvelles combinaisons et nouveaux noms nécessaires suite à la mise au point du référentiel des noms de champignons présents sur le territoire national métropolitain (1—Basidiomycètes) Doc. Mycol. 34(135–136):48–52 |
| 2008 | 17 | Kerrigan RW, Callac P, Parra LA (2008) New and rare taxa in Agaricus section Bivelares (Duploannulati). Mycologia 100:876–892. |
| 2008 | 18 | Parra LA (2008) Agaricus L., Allopsalliota, Nauta & Bas. Fungi Europaei 1. Candusso Edizioni s.a.s. Alassio, Italy. |
| 2009 | 19 | Meijer AAR de (2009) Notable macrofungi from Brazil's Paraná pine forests (Macrofungos notáveis das florestas de pinheiro-do-paraná). Embrapa Florestas, Colombo, Brazil. 431 pp. |
| 2010 | 20 | Hama O, Maes E, Guissou ML, Ibrahim D, Baragé M, Parra LA, Raspé O, De Kesel A (2010) Agaricus subsaharianus, une nouvelle espèce comestible et consommée au Niger, au Burkina Faso et en Tanzanie. Cryptogamie Mycol. 31:221–234. |
| 2010 | 21 | Moreno G, Lizárraga L, Esqueda M., Coronado ML (2010) Contribution to the study of gasteroid and secotioid fungi of Chihuahua, Mexico. Mycotaxon 112:291–315. |
| 2011 | 22 | Parra LA, Mua A, Cappelli A, Callac P (2011) Agaricus biannulatus sp. nov., a new species of the section Xanthodermatei collected in Sardinia and Sicily. Micol. Veg. Medit. 26:3–20. |
| 2012 | 23 | Chen J, Zhao RL, Karunarathna SC, Callac P, Raspé O, Bahkali AL, Hyde KD (2012) Agaricus megalosporus: a new species in section Minores. Cryptogamie Mycol. 33:145–155. |
| 2012 | 24 | Lebel T (2013) Two new species of sequestrate Agaricus (section Minores) from Australia. Mycol. Prog. 12:699–707. |
| 2012 | 25 | Lebel T, Syme A (2012) Sequestrate species of Agaricus and Macrolepiota from Australia: new species and combinations and their position in a calibrated phylogeny. Mycologia 104:496–520. |
| 2012 | 26 | Zhao RL, Desjardin DE, Callac P, Parra LA, Guinberteau J, Soytong K, Karunarathna S, Zhang Y, Hyde KD (2012a) Two species of Agaricus sect. Xanthodermatei from Thailand. Mycotaxon 122:187–195. |

*(Continued)*

**Table 2.** (Continued)

| | | |
|---|---|---|
| 2012 | 27 | Zhao RL, Hyde KD, Desjardin DE, Raspe O, Soytong K, Guinberteau J, Karunarathna SC, Callac P (2012b) Agaricus flocculosipes sp. nov., a new potentially cultivatable species from the palaeotropics. Mycoscience 53:300–311. |
| 2013 | 28 | Parra LA (2013) Agaricus L., Allopsalliota, Nauta & Bas. Fungi Europaei 1A. Candusso Edizioni s.a.s. Alassio, Italy. |
| 2014 | 29 | Karunarathna SC, Guinberteau J, Chen J, Vellinga EC, Zhao R, Chukeatirote E, Yan J, Hyde KD, Callac P (2014) Two new species in Agaricus tropical clade I. Chiang Mai J. Sci. 41:771–780. |
| 2014 | 30 | Li SF, Xi YL, Qi CX, Liang QQ, Wei SL, Li GJ, Zhao D, Li SJ, Wen HA (2014) Agaricus taeniatus sp. nov., a new member of Agaricus sect. Bivelares from northwest China. Mycotaxon 129:187–196. |
| 2014 | 31 | Parra LA, Muñoz G, Callac P (2014) Agaricus caballeroi sp. nov., una nueva especie de la sección Nigrobrunnescentes recolectada en España. Micol. Veg. Medit. 29:21–38. |
| 2014 | 32 | Thongklang N, Nawaz R, Khalid AN, Chen J, Hyde KD, Zhao RL, Parra LA, Hanif M, Moinard M, Callac P (2014a) Morphological and molecular characterization of three Agaricus species from tropical Asia (Pakistan, Thailand) reveals a new group in section Xanthodermatei. Mycologia 106:1220–1232. |
| 2015 | 33 | Gui Y, Zhu GS, Callac P, Hyde KD, Parra LA, Chen J, Yang TJ, Huang WB, Gong GL, Liu ZY (2015) Agaricus section Arvenses: three new species in highland subtropical Southwest China. Fungal Biol. 119:79–94. |
| 2015 | 34 | Liu JK, Hyde KD, Gareth EBG, Ariyawansa HA, Bhat DJ, Boonmee S, Maharachchikumbura S, McKenzie EHC, Phookamsak R, Phukhamsakda R, Abdel-Wahab MA, Buyck B, Chen J, Chethana KWT, Singtripop C, Dai DQ, Dai YC, Daranagama DA, Dissanayake AJ, Doilom M, D'souza MJ, Fan LX, Goonasekara ID, Hirayama K, Hongsanan S, Jayasiri SC, Jayawardena RS, Karunarathna SC, Li WJ, Mapook A, Norphanphoun C, Pang KL, Perera RH, Peršoh D, Pinruan U, Senanayake IC, Somrithipol S, Satinee S, Tanaka K, Thambugala KM, Tian Q, Tibpromma S, Udayanga D, Wijayawardene NN, Wanasinghe D, Abdel-Aziz FA, Adamčík S, Bahkali AH, Boonyuen N, Bulgakov T, Callac P, Chomnunti P, Greiner K, Hashimoto A, Hofstetter V, Kang JC, Li XH, Liu ZY, Matumura M, Mortimer PE, Rambold R, Randrianjohany E, Sato G, Indrasutdhi VS, Verbeken A, Brackel W, Wang Y, Wen TC, Xu JC, Yan JY, Zhao RL, Camporesi E (2015) Fungal diversity notes 1–110: taxonomic and phylogenetic contributions to fungal species. Fungal Diversity 72:1–197. |
| 2015 | 35 | Chen J, Parra LA, Guelly A, Rapior S, Hyde KD, Zao RL, Callac P (2015) Agaricus section Brunneopicti: a phylogenetic reconstruction with description of four new taxa. Phytotaxa 192:145–168. |
| 2015 | 36 | He M, Zhao RL (2015) A new species of Agaricus section Minores from China. Mycology 6:182–186. |
| 2015 | 37 | Ariyawansa HA, Hyde KD, Jayasiri SC, Buyck B, Chethana KWT, Dai DQ, Dai YC, Daranagama DA, Jayawardena RS, Lücking R, Ghobad-Nejhad M, Niskanen T, Thambugala KM, Voigt K, Zhao RL, Li GJ, Doilom M, Boonmee S, Yang ZL, Cai Q, Cui Y-Y, Bahkali AH, Chen J, Cui BK, Chen JJ, Dayarathne MC, Dissanayake AJ, Ekanayaka AH, Hashimoto A, Hongsanan S, Jones EBG, Larsson E, Li WJ, Li Q-R, Liu JK, Luo ZL, Maharachchikumbura SSN, Mapook A, McKenzie EHC, Norphanphoun C, Konta S, Pang KL, Perera RH, Phookamsak R, Phukhamsakda C, Pinruan U, Randrianjohany E, Singtripop C, Tanaka K, Tian CM, Tibpromma S, Abdel-Wahab MA, Wanasinghe DN, Wijayawardene NN, Zhang J-F, Zhang H, Abdel-Aziz FA, Wedin M, Westberg M, Ammirati JF, Bulgakov TS, Lima DX, Callaghan TM, Callac P, Chang C-H, Coca LF, Dal-Forno M, Dollhofer V, Fliegerová K, Greiner K, Griffith GW, Ho H-M, Hofstetter V, Jeewon R, Kang JC, Wen T-C, Kirk PM, Kytövuori I, Lawrey JD, Xing J, Li H, Liu ZY, Liu XZ, Liimatainen K, Lumbsch HT, Matsumura M, Moncada B, Nuankaew S, Parnmen S, de Azevedo Santiago ALCM, Sommai S, Song Y, de Souza CAF, de Souza-Motta CM, Su HY, Suetrong S, Wang YW, Wei S-F, Wen TC, Yuan HS, Zhou LW, Réblová M, Fournier J, Camporesi E, Luangsa-ard JJ, Tasanathai K, Khonsanit A, Thanakitpipattana D, Somrithipol S, Diederich P, Millanes AM, Common RS, Stadler M, Yan JY, Li XH, Lee HW, Nguyen TTT, Lee HB, Battistin E, Marsico O, Vizzini A, Vila J, Ercole E, Eberhardt U, Simonini G, Wen H-A, Chen X-H, Miettinen O, Spirin V, Hernawati (2015) Fungal diversity notes 111–252 taxonomic and phylogenetic contributions to fungal taxa. Fungal Diversity 75:27–274. |
| 2015 | 38 | Parra LA, Wisman J, Guinberteau J, Wilhelm M, Weholt Ø, Musumeci E, Callac P, Geml J. (2015) Agaricus collegarum and Agaricus masoalensis, two new taxa of the section Nigrobrunnescentes collected in Europe. Micol. Veget. Medit. 30:3–26. |
| 2015 | 39 | Wang ZR, Parra LA, Callac P, Zhou JL, Fu WJ, Dui SH, Hyde KD, Zhao RL (2015) Edible species of Agaricus (Agaricaceae) from Xinjiang Province (Western China). Phytotaxa 202:185–197. |
| 2016 | 40 | Bates ST, Chapman RM, Islam MB, Schwabe A, Wardenaar ECP, Evenson VS (2016) Phylogenetic placement of the secotioid fungus Araneosa columellata within Agaricus. Mycotaxon. 131:103–110. |
| 2016 | 41 | Chen J, Parra LA, De Kesel A, Khalid AN, Qasim T, Aisha A, Bahkali AH, Hyde KD, Zhao RL, Callac P (2016) Inter–and intra–specific diversity in Agaricus endoxanthus and allied species reveals a new taxon, A. punjabensis. Phytotaxa 252:1–16. |

*(Continued)*

**Table 2.** (Continued)

| 2016 | 42 | Dai RC, Li GJ, He MQ, Liu RL, Ling ZL, Wu JR, Zhao RL (2016) Characterization of four species including one new species of Agaricus subgenus Spissicaules from Eastern China. Mycosphere 7:405–416. |
|---|---|---|
| 2016 | 43 | Kaur M, Kaur H, Malik NA (2016) Identification and characterization of six new taxa of genus Agaricus L.:FR. (Agaricaceae Chevall.) from India. World Journal of Pharmacy and Pharmaceutical Sciences 5:1518–1533. |
| 2016 | 44 | Kerrigan RW (2016) Agaricus of North America. New York Botanical Garden. |
| 2016 | 45 | Li GJ, Hyde KD, Zhao RL, Hongsanan S, Abdel-Aziz FA, Abdel-Wahab MA, Alvarado P, Alves-Silva G, Ammirati JF, Ariyawansa HA, Baghela A, Bahkali AH, Beug M, Bhat DJ, Bojantchev D, Boonpratuang T, Bulgakov TS, Camporesi E, Boro MC, Ceska O, Chakraborty D, Chen JJ, Chethana KWT, Chomnunti P, Consiglio G, Cui BK, Dai DQ, Dai YC, Daranagama DA, Das K, Dayarathne MC, De Crop E, De Oliveira RJV, de Souza CAF, de Souza JI, Dentinger BTM, Dissanayake AJ, Doilom M, Drechsler-Santos ER, Ghobad-Nejhad M, Gilmore SP, Góes-Neto A, Gorczak M, Haitjema CH, Hapuarachchi KK, Hashimoto A, He MQ, Henske JK, Hirayama K, Iribarren MJ, Jayasiri SC, Jayawardena RS, Jeon SJ, Jerônimo GH, Jesus AL, Jones EBG, Kang JC, Karunarathna SC, Kirk PM, Konta S, Kuhnert E, Lagner E, Lee HS, Lee HB, Li WJ, Li XH, Liimatainen K, Lima DX, Lin CG, Liu JK, Liu XZ, Liu ZY, Luangsa-ard JJ, Lücking R, Lumbsch HT, Lumyong S, Leaño EM, Marano AV, Matsumura M, McKenzie EHC, Mongkolsamrit S, Mortimer PE, Nguyen TTT, Niskanen T, Norphanphoun C, O'Malley MA, Parnmen S, Pawlowska J, Perera RH, Phookamsak R, Phukhamsakda C, Pires-Zottarelli CLA, Raspé O, Reck MA, Rocha SCO, de Santiago ALCMA, Senanayake IC, Setti L, Shang QJ, Singh SK, Sir EB, Solomon KV, Song J, Sriktikulchai P, Stadler M, Suetrong S, Takahashi H, Takahashi T, Tanaka K, Tang LP, Thambugala KM, Thanakitpipattana D, Theodorou MK, Thongbai B, Thummarukcharoen T, Tian Q, Tibpromma S, Verbeken A, Vizzini A, Vlasák J, Voigt K, Wanasinghe DN, Wang Y, Weerakoon G, Wen HA, Wen TC, Wijayawardena NN, Wongkanoun S, Wrzosek M, Xiao YP, Xu JC, Yan JY, Yang J, Yang SD, Hu Y, Zhang JF, Zhao J, Zhou LW, Peršoh D, Phillips AJL, Maharachchikumbura SSN. (2016) Fungal diversity notes 253–366: taxonomic and phylogenetic contributions to fungal taxa. Fungal Diversity 78:1–237. |
| 2016 | 46 | Thongklang N, Chen J, Bandara AR, Hyde KD, Raspé O, Parra LA, Callac P (2016) Studies on Agaricus subtilipes, a new cultivatable species from Thailand, incidentally reveal the presence of Agaricus subrufescens in Africa. Mycoscience 57:239–250. |
| 2016 | 47 | Zhao RL, Zhou JL, Chen J, Margaritescu S, Sánchez-Ramírez S, Hyde KD, Callac P, Parra LA, Li GJ, Moncalvo JM (2016) Towards standardizing taxonomic ranks using divergence times–a case study for reconstruction of the Agaricus taxonomic system. Fungal diversity 78:239–292. |
| 2016 | 48 | Zhou JL, Su SY, Su HY, Wang B, Callac P, Guinberteau J, Hyde KD, Zhao RL (2016) Description of eleven new species of Agaricus sections Xanthodermatei and Hondenses collected from Tibet and the surrounding areas. Phytotaxa 257:99–121. |
| 2017 | 49 | Chen J, Callac P, Parra LA, Karunarathna SC, He MQ, Moinard M, De Kesel A, Raspé O, Wisitrassameewong K, Hyde KD, Zhao RL (2017) Study in Agaricus subgenus Minores and allied clades reveals a new American subgenus and contrasting phylogenetic patterns in Europe and Greater Mekong Subregion. Persoonia 38:170–196. |
| 2017 | 50 | Drewinski M de P, Junior NM, Neves MA (2017) Agaricus globocystidiatus: a new neotropical species with pleurocystidia in Agaricus subg. Minoriopsis. Phytotaxa 314:64–72. |
| 2017 | 51 | He MQ, Chen J, Zhou JL, Ratchadawan C, Hyde KD, Zhao, RL (2017) Tropic origins, a dispersal model for saprotrophic mushrooms in Agaricus section Minores with descriptions of sixteen new species. Scientific reports 7:5122. |
| 2017 | 52 | Hyde KD, Norphanphoun C, Abreu VP, Bazzicalupo A, Chethana KWT, Clericuzio M, Dayarathne MC, Dissanayake AJ, Ekanayaka AH, He MQ, Hongsanan S, Huang SK, Jayasiri SC, Jayawardena RS, Karunarathna A, Konta S, Kušan I, Lee H, Li JF, Lin CG, Liu NG, Lu YZ, Luo ZL, Manawasinghe IS, Mapook A, Perera RH, Phookamsak R, Phukhamsakda C, Siedlecki I, Mayra Soares A, Tennakoon DS, Tian Q, Tibpromma S, Wanasinghe DN, Xiao YP, Yang J, Zeng XY, Abdel-Aziz FA, Li WJ, Senanayake IC, Shang QJ, Daranagama DA, de Silva NI, Thambugala KM, Abdel-Wahab MA, Bahkali AH, Berbee ML, Boonmee S, Bhat DJ, Bulgakov TS, Buyck B, Camporesi E, Castañeda-Ruiz RF, Chomnunti P, Doilom M, Dovana F, Gibertoni TB, Jadan M, Jeewon R, Jones EBG, Kang JC, Karunarathna SC, Lim YW, Liu JK, Liu ZY, Plautz HL Jr, Lumyong S, Maharachchikumbura SSN, Matočec N, McKenzie EHC, Mešić A, Miller D, Pawłowska J, Pereira OL, Promputtha I, Romero AI, Ryvarden L, Su HY, Suetrong S, Tkalčec Z, Vizzini A, Wen TC, Wisitrassameewong K, Wrzosek M, Xu JC, Zhao Q, Zhao RL, Mortimer PE (2017) Fungal diversity notes 603–708: taxonomic and phylogenetic notes on genera and species. Fungal Diversity 87:1–235. |
| 2017 | 53 | Mua A, Casula M, Sanna M (2017) Agaricus ornatipes sp. nov., una novua species della sezione Arvenses raccolta in Sardegna. Micol. Veget. Medit. 32:59–74. |

(*Continued*)

Table 2. (Continued)

| 2017 | 54 | Parra LA, Caballero A (2017) Agaricus pietatis, una especie nueva de Agaricus sect. Minores encontrada en España. Boletín Micológico de FAMCAL 12:137–143. |
|---|---|---|
| 2017 | 55 | Zhang MZ, Li GJ, Dai RC, Xi YL, Wei SL, Zhao RL (2017) The edible wide mushrooms of Agaricus section Bivelares from Western China. Mycosphere 8:1640–1652. |
| 2018 | 56 | Bashir H, Hussain S, Khalid AN, Khan Niazi AR, Parra L, Callac P (2018) First report of Agaricus sect. Brunneopicti from Pakistan with descriptions of two new species. Phytotaxa 357:167–178. |
| 2018 | 57 | He MQ, Chuankid B, Hyde KD, Cheewangkoon R, Zhao RL (2018a) A new section and species of Agaricus subgenus Pseudochitonia from Thailand. MycoKeys 40:53–67. |
| 2018 | 58 | He MQ, Hyde KD, Wei SL, Xi YL, Cheewangkoon R, Zhao RL (2018b) Three new species of Agaricus section Minores from China. Mycosphere 9:189–201. |
| 2018 | 59 | Mahdizadeh V, Parra LA, Safaie N, Goltapeh EM, Chen J, Guinberteau J, Callac P (2018) A phylogenetic and morphological overview of sections Bohusia, Sanguinolenti, and allied sections within Agaricus subg. Pseudochitonia with three new species from France, Iran and Portugal. Fungal Biol. 112:34–53. |
| 2018 | 60 | Parra LA, Angelini C, Ortiz-Santana B, Mata G, Billette C, Rojo C, Chen J, Callac P (2018) The genus Agaricus in the Caribbean. Nine new taxa mostly based on collections from the Dominican Republic. Phytotaxa 345:219–271. |
| 2018 | 61 | Tarafder E, Dutta AK, Sarkar J, Acharya K (2018) A new species of Agaricus sect. Brunneopicti from Eastern India. Phytotaxa 374: 139–146. |
| 2019 | 62 | Parra LA, Cappelli A, Kerrigan RW, Bizio E 2019 [2018] *Agaricus porphyrocephalus* subsp. *alpinus* a new subspecies collected in the Italian Alps Micol. Veg. Medit. 33(2):67–88 |
| 2019 | 63 | Phookamsak R, Hyde KD, Jeewon R, Bhat DJ, Jones EBG, Maharachchikumbura SSN, Raspé O, Karunarathna SC, Wanasinghe DN, Hongsanan S, Doilom M, Tennakoon DS, Machado AR, Firmino AL, Ghosh A, Karunarathna A, Mešić A, Dutta AK, Thongbai B, Devadatha B, Norphanphoun C, Senwanna C, Wei D, Pem D, Ackah FK, Wang G-N, Jiang HB, Madrid H, Lee HB, Goonasekara ID, Manawasinghe IS, Kušan I, Cano J, Gené J, Li J, Das K, Acharya K, Raj KNA, Latha KPD, Chethana KWT, He MQ, Dueñas M, Jadan M, Martín MP, Samarakoon MC, Dayarathne MC, Raza M, Park MS, Telleria MT, Chaiwan N, Matočec N, de Silva NI, Pereira OL, Singh PN, Manimohan P, Uniyal P, Shang QJ, Bhatt RP, Perera RH, Alvarenga RLM, Nogal-Prata S, Singh SK, Vadthanarat S, Oh SY, Huang SK, Rana S, Konta S, Paloi S, Jayasiri SC, Jeon S J, Mehmood T, Gibertoni TB, Nguyen TTT, Singh U, Thiyagaraja V, Sarma VV, Dong W, Yu XD, Lu YZ, Lim YW, Chen Y, Tkalčec Z, Zhang ZF, Luo ZL, Daranagama DA, Thambugala KM, Tibpromma S, Camporesi E, Bulgakov TS, Dissanayake AJ, Senanayake IC, Dai DQ, Tang LZ, Khan S, Zhang H, Promputtha I, Cai L, Chomnunti P, Zhao RL, Lumyong S, Boonmee S, Wen TC, Mortimer PE, Xu JC (2019) Fungal diversity notes 929–1035: taxonomic and phylogenetic contributions on genera and species of fungi. Fungal Diversity 95:1–273. |
| 2019 | 64 | Chen J, Callac P, Llarena Hernandez RC, Mata G (2019) Two species of Agaricus subg. Minoriopsis from Mexico. Phytotaxa 404(3):091–101. |
| 2019 | 65 | Hussain S, Sher H. (2019) Study in Agaricus section Minores in Pakistan with the description of two new species. Mycological Progress 18:795–204. |
| 2019 | 66 | Zheng J, Li J., Song Y, Wang G, Qiu L (2019) Agaricus rubripes sp. nov., a new species from Southern China. Nova Hedwigia 109(1–2):233–246. |
| 2020 | 67 | Liu AQ, Dai RC, Zhang MZ, Cao B, Xi YL, Wei SL, Zhao RL (2020) Species of *Agaricus* section *Agaricus* from China. Phytotaxa 452(1):1–18. |
| 2020 | 68 | Cao B, He MQ, Ling ZL, Zhang MZ, Wei SL, Zhao RL. A revision of *Agaricus* section *Arvenses* with nine new species from China. Mycologia. 2020 Dec 16:1–21. doi: 10.1080/00275514.2020.1830247. Epub ahead of print. PMID: 33326360. |
| 2020 | 69 | Mua A., Porcu G., Casula M., Sanna M. (2020) *Agaricus pixinortui* sp. nov., una nouva specie della sezione Nigrobrunnescentes raccolta in Sardegna. Micol. Veget. Medit. 35(1):3–12. |
| 2021 | 70 | Ling Z.-L., Zhou J.-L., Parra L.A., De Kesel A., Callac P., Cao B., He M.-Q., Zhao R.-L. (2021) Four new species of *Agaricus* subgenus *Spissicaules* from China. Mycologia 113:2, 476–491. https://doi.org/10.1080/00275514.2020.1852808 |
| 2021 | 71 | Ortiz-Santana B., Chen J., Parra L.A., Angelini C., Lodge D.J., Kerrigan R.W., Callac P. (2021) The genus *Agaricus* in the Caribbean II. Refined phylogeny of *Agaricus* subg. *Spissicaules* with description of two new sections and eight new species. Mycological Progress 20, 381–411. https://doi.org/10.1007/s11557-021-01686-9 |
| 2021 | 72 | Parra LA, Cappelli A, Benazza-Bouregba M (2021) On two rare species of *A*. subsect. *Pattersonia*: *Agaricus salvatoris* sp. nov. and new records of *A. lapparrae* from Europe and Africa. Micol. Veget. Medit. (2020), 35 (2): 91–112. |

*(Continued)*

**Table 2.** (Continued)

| Year | # | Reference |
|---|---|---|
| 2021 | 73 | Bashir H, Chen J, Jabeen S, Ullah S, Khan J5, Niazi AR, Zhang M, Khalid AN, Parra LA and Callac P (2021) An overview of Agaricus section Hondenses and Agaricus section Xanthodermatei with description of eight new species from Pakistan. Scientific Reports: 11, 12905 (https://doi.org/10.1038/s41598-021-92261-5) |
| 2021 | 74 | Jaichaliaw C, Kumla J, Vadthanarat S, Suwannarach N, Lumyong S (2021) Multigene Phylogeny and Morphology Reveal Three Novel Species and a Novel Record of Agaricus from Northern Thailand. Frontiers in Microbiology 12:650513. doi: 10.3389/fmicb.2021.650513 |
| 2021 | 75 | Blanco-Dios J.B. (2021) Notas nomenclaturales en los Órdenes Agaricales, Boletales, Cantharellales, Polyporales y Russulales. Yesca, Revista Sociedad Micológica de Cántabria 33: 105–111. |
| 2021 | 76 | Parra L.A., Faraoni M., Suriano E. (2021) Agaricus carassaii una nuvea especie de Agaricus sect. Minores recolectada en Italia central Micol. Veg. Medit. 36(1–2):3–22 |
| 2022 | 77 | Ferretti A., Saar I., Knijn A. (2022). A new species of Agaricus (section Sanguinolenti) from Rome, Italy. Italian Journal of Mycology 51:1–10 |
| 2022 | 78 | Arya C. P., Manoj Kumar A., Pradeep C. K., Parra L. A. (2022) *Agaricus brunneodiscus*, a new speccies of *Agaricus* section *Rarolentes* from India. Phytotaxa 533(4):181–193. |
| 2022 | 79 | Tarafder E., Dutta A.K., Acharya K. (2022) New species and new record in *Agaricus* subg. *Minores* from India. Turkish Journal of Botany **46:183–195** doi:10.3906/bot-2111-62 |
| 2022 | 80 | Medel-Ortiz R., Garibay-Orijel R., Argüelles-Moyao A., Mata G., Kerrigan R.W., Bessette A.E., Geml J., Angelini, C., Parra L.A., Chen, J. (2022) *Agaricus macrochlamys*, a new species from the (sub)tropical cloud forests of North America and the Caribbean, and *Agaricus fiardii*, a new synonym of *Agaricus subrufescens*. J. Fungi 8:664. https://doi.org/10.3390/jof8070664 |
| 2022 | 81 | Crous, P.W.; Boers, J.; Holdom, D.; Osieck, E.R.; Steinrucken, T.V.; Tan, Y.P.; Vitelli, J.S.; Shivas, R.G.; Barrett, M.; Boxshall, A.-G.; Broadbridge, J.; Larsson, E.; Lebel, T.; Pinruan, U.; Sommai, S.; Alvarado, P.; Bonito, G.; Decock, C.A.; De la Peña-Lastra, S.; Delgado, G.; Houbraken, J.; Maciá-Vicente, J.G.; Raja, H. A.; Rigueiro-Rodríguez, A.; Rodríguez, A.; Wingfield, M.J.; Adams, S.J.; Akulov, A.; AL-Hidmi, T.; Antonín, V.; Arauzo, S.; Arenas, F.; Armada, F.; Aylward, J.; Bellanger, J.-M.; Berraf-Tebbal, A.; Bidaud, A.; Boccardo, F.; Cabero, J.; Calledda, F.; Corriol, G.; Crane, J.L.; Dearnaley, J.D.W.; Dima, B.; Dovana, F.; Eichmeier, A.; Esteve-Raventós, F.; Fine, M.; Ganzert, L.; García, D.; Torres-Garcia, D.; Gené, J.; Gutiérrez, A.; Iglesias, P.; Istel, Ł.; Jangsantear, P.; Jansen, G.M.; Jeppson, M.; Karun, N.C.; Karich, A.; Khamsuntorn, P.; Kokkonen, K.; Kolařík, M.; Kubátová, A.; Labuda, R.; Lagashetti, A.C.; Lifshitz, N.; Linde, C.; Loizides, M.; Luangsa-ard, J.J.; Lueangjaroenkit, P.; Mahadevakumar, S.; Mahamedi, A.E.; Malloch, D.W.; Marincowitz, S.; Mateos, A.; Moreau, P.-A.; Miller, A.N.; Molia, A.; Morte, A.; Navarro-Ródenas, A.; Nebesářová, J.; Nigrone, E.; Nuthan, B.R.; Oberlies, N.H.; Pepori, A.L.; Rämä, T.; Rapley, D.; Reschke, K.; Robicheau, B.M.; Roets, F.; Roux, J.; Saavedra, M.; Sakolrak, B.; Santini, A.; Ševčíková, H.; Singh, P.N.; Singh, S.K.; Somrithipol, S.; Spetik, M.; Sridhar, K.R.; Starink-Willemse, M.; Taylor, V.A.; Iperen, A.L. van; Vauras, J.; Walker, A.K.; Wingfield, B.D.; Yarden, O.; Cooke, A.W.; Manners, A.G.; Pegg, K.G.; Groenewald, J.Z.(2022) Fungal planets description sheets: 1383–1435. Persoonia 48: 261–371. https://doi.org/10.3767/persoonia.2022.48.08 |
| 2022 | 82 | Hussain, S., Al-Kharousi, M., Al-Muharabi, M.A., Al-Maqbali, D., Zahra Al-Shabibi, Z., Al-Balushi, A. H., Al-Yahya'ei, M. N., Al Saady, N., Abdel-Jalil, R., Velazhahan, R., Al-Sadi, A. M. (2022) Phylogeny of Agaricus subgenus Pseudochitonia with the description of a new section and a new species from Oman. Mycol Progress 21, 72. https://doi.org/10.1007/s11557-022-01819-8 |
| 2022 | 83 | Cappelli A., Parra Sanchez L;A. (2022) Agaricus L. from European Mediterranean countries. Fungi non delineati, parts 76/78. Candusso Editrice, Orrigio, Italy. Pp. 520. |
| 2022 | 84 | Index Fungorum Kerrigan R.W., Parra L.A. Nomenclatural novelties n˚ 526: 1 |
| 2022 | 85 | Arya C.P., Pradeep C.K. Pradeep (2022) A new species of Agaricus sect. Agaricus (Agaricaceae) from India. Nordic Journal of Botany: e03742:1–7. doi: 10.1111/njb.03742 |
| **25 references (n˚61 to 85) with new taxa since the paper of Callac & Chen (2018)** | | |
| 2003 | A | Challen M. P., Kerrigan R. W. et Callac P. (2003) A phylogenetic reconstruction and emendation of Agaricus section Duploannulatae, Mycologia 95: 61–71. http://dx.doi.org/10.1080/15572536.2004.11833132 |
| 2006 | B | Kerrigan RW., Callac P., Guinberteau J., Challen M. et Parra LA. (2006). *Agaricus* section *Xanthodermatei*: a phylogenetic reconstruction with commentary on taxa. **Mycologia** 97:1292–1315 {« 2005 »]. |
| 2011 | C | Zhao R., Karunarathna S., Raspé O., Parra L. A., Guinberteau J., Moinard M., De Kesel A., Barroso G., Courtecuisse R., Hyde K.D., Guelly A.K., Desjardin D.E., Callac P. (2011). Major clades in tropical Agaricus. Fungal Diversity 51: 279–296. https://doi.org/10.1007/s13225-011-0136-7 |
| 2016 | D | Mahdizadeh V., Safaie N., Goltapeh E.M., Asef M.R., Nassaj Hosseini S.M, Callac P. (2016). Agaricus section Xanthodermatei in Iran. Phytotaxa 247:181–196. http://dx.doi.org/10.11646/phytotaxa.247.3.2 |

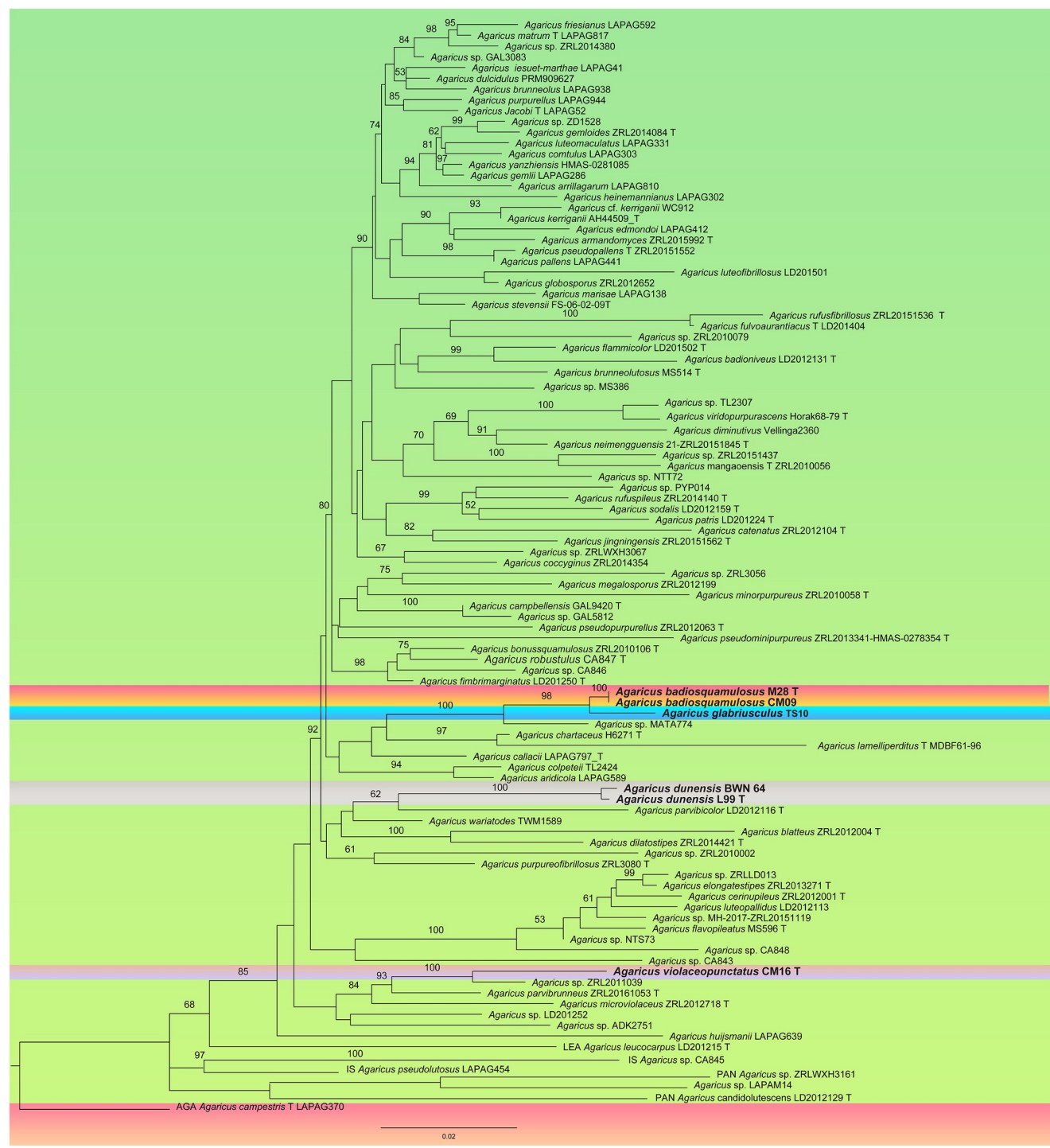

**Fig 2. Maximum likelihood tree of species belonging to A. sect. Minores generated from combined LSU, ITS and TEF1 sequence data.** Agaricus campestris is used as outgroup. Bootstrap support values are indicated (BS). New species from Pakistan are in bold red. T = type specimen.

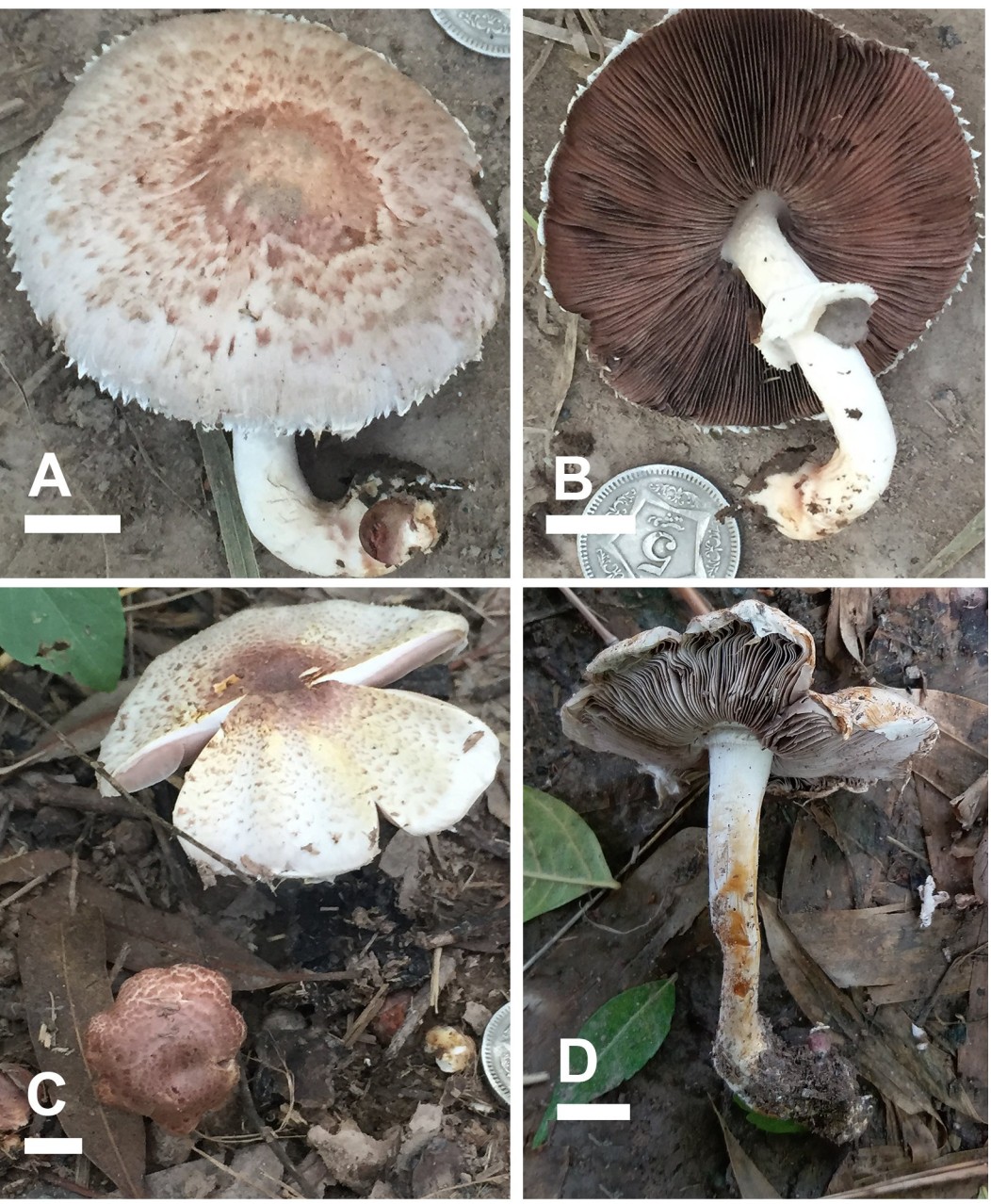

**Fig 3. Macromorphological characters of *A. badiosquamulosus* (A–D) basidiomata in the field.** Bars = 1 cm. Photographed by Dr. Hira Bashir.

**Habitat and distribution:** Gregarious on grassy grounds, some under *Eucalyptus camaldulensis* Dehnh. trees along Pak. Motorway (M2 & M3) and TS57 found near *Dalbergia sissoo* trees along roadside in sandy soil. It is only known from Punjab province of Pakistan so far.

**Additional material examined:** PAKISTAN, Punjab, Lahore, solitary on rich loamy soil along Pak. Motorway (M3) near *Eucalyptus camadulensis*, 217 m a.s.l., 14 August 2016, Hira Bashir, **M4** (LAH35752), (GenBank # ON137222); Punjab, Lahore, solitary on rich loamy soil along Pak. Motorway (M3) near *Eucalyptus camadulensis*, 217 m a.s.l., 14 August 2016, Hira

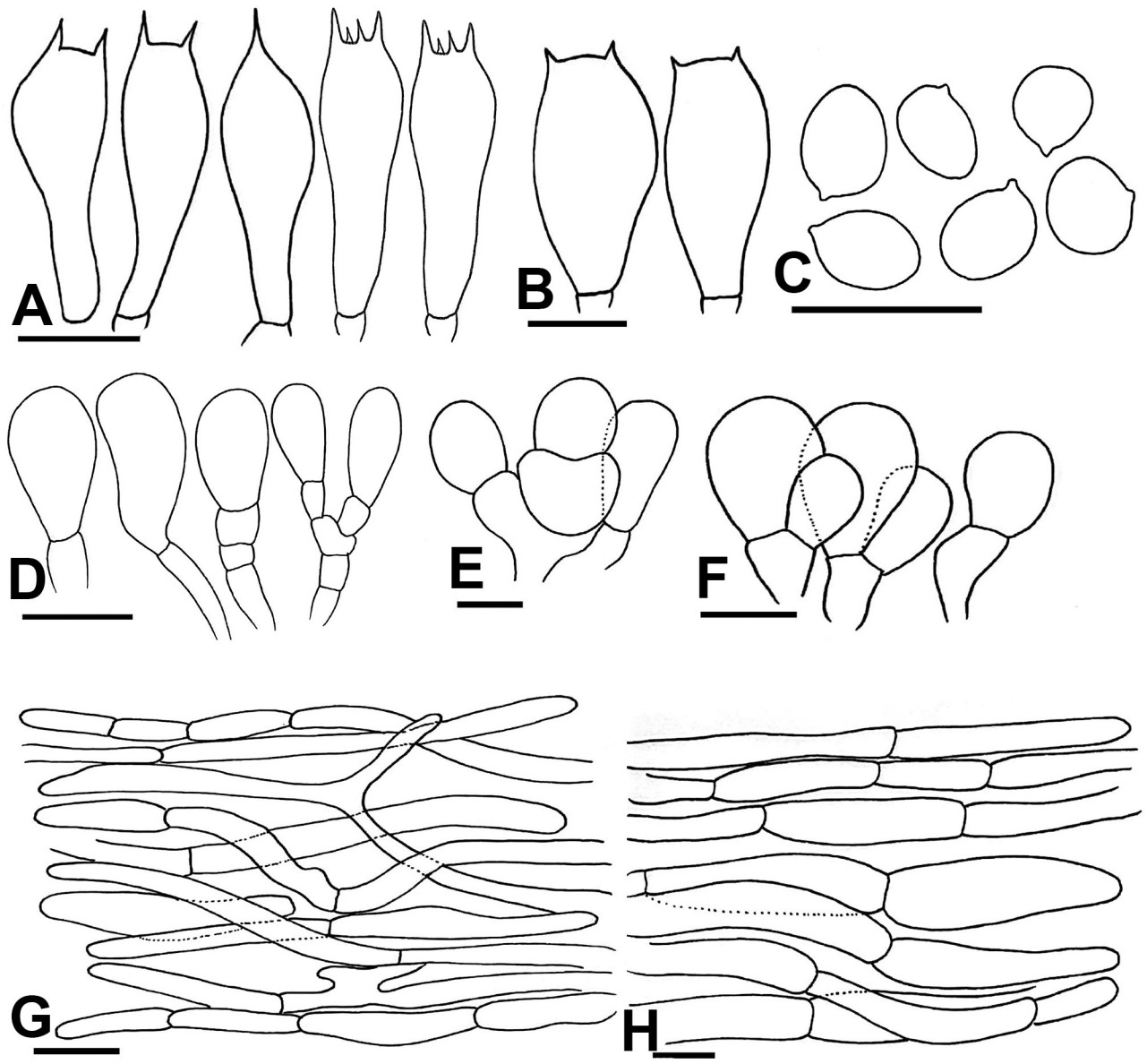

**Fig 4. Micromorphological characters of A. badiosquamulosus.** (A–H); (A-B) Basidia, (C) Basidiospores, (D-F) Cheilocystidia, (G) Pileipellis hyphae, (H) Stipitipellis hyphae. Bars = 10 μm. Drawings by Dr. Hira Bashir.

Bashir, **M6** (LAH35753), (GenBank # ON137223); Punjab, Lahore, solitary on rich loamy soil along Pak. Motorway (M3) under *Eucalyptus camadulensis*, 217 m a.s.l., 1 September 2016, Hira Bashir, **M22** (LAH35754), (GenBank # ON137224); Punjab, Lahore, solitary on rich loamy soil along motorway *Eucalyptus camadulensis*, 217 m a.s.l., 4 August 2015, Hira Bashir, **F3** (LAH35755), (GenBank # ON137225); Punjab, Lahore, solitary on rich loamy soil along Pak. Motorway (M2) under *Eucalyptus camadulensis*, 217 m a.s.l., 4 August 2015, Hira Bashir, **F4** (LAH35756), (GenBank # ON137226); Punjab, Lahore, solitary on rich loamy soil along Pak. Motorway (M2) near *Eucalyptus camadulensis*, 217 m a.s.l., 4 August 2015, Hira Bashir, **F5** (LAH35757), (GenBank # ON137227); Punjab, Lahore, solitary on rich loamy soil along

**Table 3. Polymorphic positions within different samples of closely related species *A. badiosquamulosus*, and *A. glabriusculus*.**

| Samples | Heteromorphic Positions | | | | | | | | | | | | | | | | | | | | |
|---|---|---|---|---|---|---|---|---|---|---|---|---|---|---|---|---|---|---|---|---|---|
| | 9 | 10 | 11 | 12 | 13 | 127 | 222 | 226 | 279 | 298 | 306 | 314 | 440 | 522 | 535 | 571 | 651 | 669 | 696 | 701 | 720 |
| **TS2** *A. glabriusculus* | - | - | - | - | T | T | C | G | A | C | T | T | T | A | C | A | A | - | C | T | C |
| **TS10** | - | - | - | - | T | K | C | G | A | C | T | W | Y | R | C | A | M | - | S | T | C |
| **TS11** | - | - | - | - | T | K | C | G | A | C | T | W | Y | R | C | A | M | - | S | T | C |
| SH7 *A. glabriusculus* | T | T | T | A | T | T | C | G | A | C | T | T | T | A | C | A | A | - | G | T | T |
| SH 291L | T | T | T | A | T | T | C | G | A | C | T | T | T | A | C | A | A | - | G | T | T |
| **M4** *A. badiosquamulosus* | - | - | - | - | T | T | C | G | A | T | C | A | T | G | T | G | A | T | C | G | C |
| **TS57** | - | - | - | - | T | T | C | G | R | T | C | A | T | G | T | G | A | T | C | G | C |
| **F11** | - | - | - | - | T | T | R | G | A | T | C | A | T | G | T | G | A | T | C | T | C |
| **F5** | - | - | - | - | T | T | C | R | A | T | C | A | T | G | T | G | A | T | C | T | C |
| **F4** | - | - | - | - | T | T | C | R | A | T | C | A | T | G | T | G | A | T | C | T | C |
| **F3** | - | - | - | - | T | T | C | R | A | T | C | A | T | G | T | G | A | T | C | T | C |
| **M22** | - | - | - | - | T | T | C | R | A | T | C | A | T | G | T | G | A | T | C | T | C |
| **M6** | - | - | - | - | W | T | C | G | A | T | C | A | T | G | T | G | A | T | C | T | C |
| **M28** | - | - | - | - | T | T | C | G | A | T | C | A | T | G | T | G | A | T | C | T | C |
| **F53** | - | - | - | - | T | T | C | G | A | T | C | A | T | G | T | G | A | T | C | T | C |
| **CM09** | - | - | - | - | T | T | C | G | A | T | C | A | T | G | T | G | A | T | C | T | C |

Heteromorphisms are shown by yellow highlight. Newly added samples are indicated in bold.

Pak. Motorway (M2) near *Eucalyptus camadulensis*, 217 m a.s.l., 4 August 2015, Hira Bashir, **F11** (LAH35758), (GenBank # ON137228); Punjab, Lahore, gregarious on rich loamy soil along Pak. Motorway (M2) near under *Eucalyptus camadulensis*, 217 m a.s.l., 6 August 2015, Hira Bashir, **F53** (LAH35759), (GenBank # ON137229); Punjab, Toba Tek Singh, at 183 m a.s.l., solitary in grassy ground near *Dalbergia sissoo*, 30 June 2017, Hira Bashir, **TS57**, (LAH35760), (GenBank # ON137230). Punjab, Kasur district, Changa Manga forest, scattered on decomposed matter under deciduous trees, 218 m a.s.l., 2 September 2021, Aneeqa Ghaffor and A.R. Niazi, **CM09** (LAH02921), (GenBank # ON137231).

**Notes**:

Phylogenetic analyses revealed that *A. badiosquamulosus* and *A. glabriusculus* clustered close to *A. glabriusculus* (SH7, SH291L) reported from Pakistan also [17]. There are 12 nucleotide differences of *A. glabriusculus* from *A. badiosquamulosus* excluding heteromorphic sites. The other three taxa form sister subclade with *A. purpureosquamulosus* (CUH AM716), *A. goossensiae* (GF929) and an undescribed *Agaricus* sp. (MATA774) within *A.* sect. *Minores*. In the ITS sequences of *A. badiosquamulosus* and *A. glabriusculus*, numerous heteromorphisms were observed in the samples (Table 3) and a nucleotide *G* at position 701 is present only in two samples (M4 and TS57) instead of *T*. A single nucleotide difference is not considered very significant in *Agaricus*, especially in case of species having numerous polymorphic positions. Also, the morphological characters strongly constrained the phylogenetic results. So, these two specimens are considered to be the same as *A. badiosquamulosus*. *Agaricus glabriusculus* has five unique sites when compared with *A. badiosquamulosus*. As far as the *A.* sp. (MATA774) is concerned, there are 18 nucleotide differences from *A. badiosquamulosus* and 19 from *A. glabriusculus* excluding heteromorphic sites. The data indicate that our species *A. badiosquamulosus* is unique from all other taxa.

***Agaricus dunensis* H. Bashir & M. Asif** (Figs 5–7)

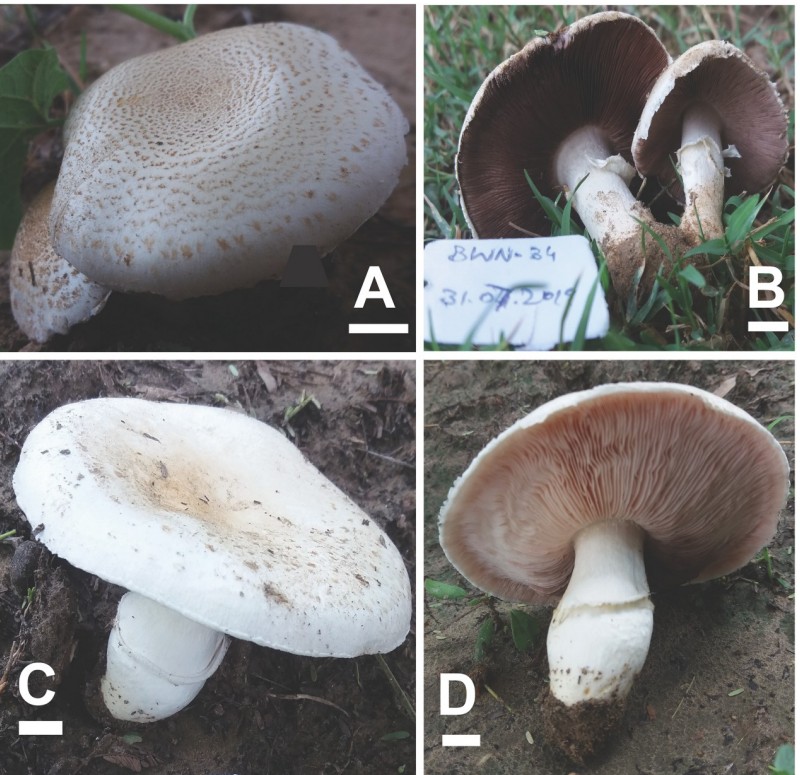

**Fig 5.** Macromorphological characters of *A. dunensis* [BWN-34 (A-B), BWN-64 (C-D)] collected from Bahawalnagar. (A–D) basidiomata in the field. Bars = 1 cm. Photographed by Dr. Hira Bashir and Muhammad Asif.

**Etymology:** '*dunensis*' in Latin referred to sand dune area from where the type specimen was recorded for the first time.

**Holotype:** Punjab, Lal Suhanra National Park, Cholistan desert, Bahawalpur, at 140 m a.s.l., scattered on sandy soil under *Acacia nilotica* trees, 23 June 2017, Hira Bashir, **L99**, (LAH 35747), GenBank # ON137217 (nrITS).

**Diagnosis:** This species is distinguished well having a pileus with bright yellow to rust squamules becoming brownish at maturity, pileus and stipe surface immediately turning dark brown when touched, and a strong red to blackish–brown discoloration upon bruising, monosporic basidia frequently observed in one sample (L2), frequently bisporic to tetrasporic in others, cheilocystidia polymorphous.

**Description: Pileus** 2.5–9 cm in diam., convex then plano-convex, slightly appressed and flattened at disc (LS4), sometimes disc depressed (BWN-64) at maturity, thoroughly covered by orange-yellow (5.8YR 5.9/6.2) squamules (predominant in L99 collection) in young stages, at maturity surface has dotted bright yellow (2.1Y 4.1/5.4) squamules becoming yellowish brown (10YR6/8) in some fruiting bodies, few scattered towards margin more dense at the center in mature sporocarp on a creamy (3.1G 8.5/0.7) background. **Surface** dull and dry. **Margins** slightly appendiculate, white, slightly incurved when young become straight at maturity, slightly exceeding lamellae. **Lamellae** light pink (7.2R 7.5/2) at young stage to dark brown (5.7R 2.2/1.4) at maturity, free, crowded, intercalated with lamellulae, entire edges sometimes slightly wavy. **Stipe** 1.5–6.5 × 0.5–3 cm, cylindrical to slightly bulbous at the base to ventricose with bulbous from center tapering towards base (BWN-64), stuffed, provided with an annulus

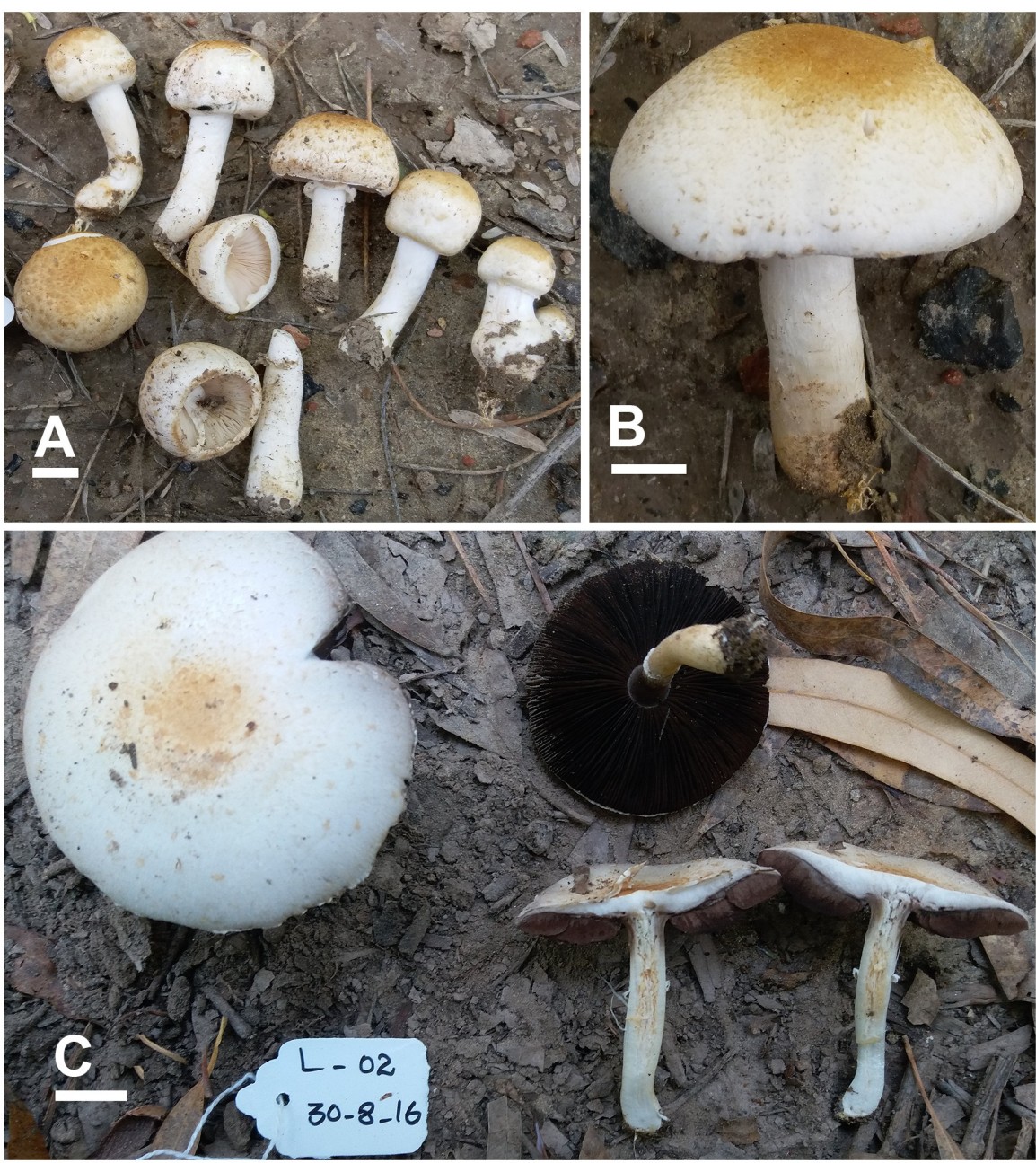

**Fig 6.** Macromorphological characters of *A. dunensis* [L99 (A-B), LS4 (C)] collected from Cholistan desert. (A–C) basidiomata in the field. Bars = 1 cm. Photographed by Dr. Hira Bashir.

in its upper part close to lamellae, covered with creamy white squamules from above and below annulus, creamy when young becomes brown from top at maturity, discoloring yellow or rusty at stipe base when rubbed, becoming light yellowish brown (1.3Y 6/2.4) when cut in L2, no significant discoloration observed in L99, brown (9.7YR 5.5/2.8) color appears mildly at stipe base in other specimens. **Annulus** superous, single edged, white (8.2PB 9.3/0.9), ruptured at young stage leaving behind a ring zone at maturity, fragile, smooth, membranous or thick

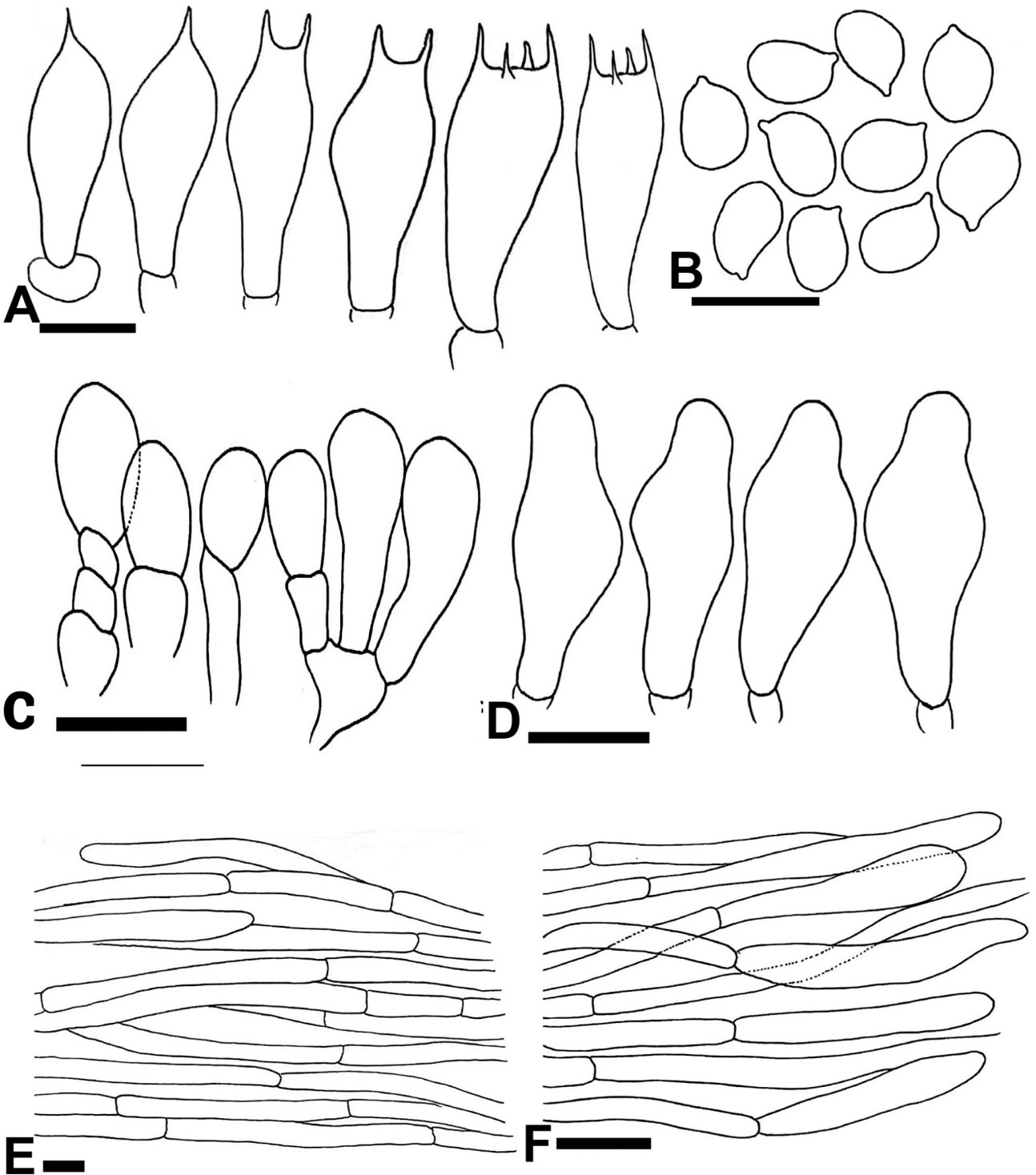

**Fig 7.** Micromorphological characters of *A. dunensis* (A–F); (A) Basidia, (B) Basidiospores, (C-D) Cheilocystidia, (E) Pileipellis hyphae, (F) Stipitipellis hyphae. Bars = 10 μm. Drawings by Dr. Hira Bashir.

and floccose on both sides (BWN-64). **Context** whitish (0.9PB 8.6/2.4), stuffed, no discoloration when cut. **Odor** strong almond-like. (Figs 5 and 6).

**Macrochemical Reactions:** KOH positive yellow. Schäffer's reaction positive, yellow then strong reddish-orange on fresh sporocarp.

**Basidiospores** (5.5–) 6.3–6.9 (–7.5) × (4.2–) 5–5.3 (–5.6) μm, [avX = 6.5 ± 0.78 × 5.2 ± 0.46 μm, $Q_m$ = 1.3, n = 20 × 9], subglobose to ellipsoid, light to dark brown in KOH, smooth with prominent apiculus, without apical pore, with granular content. **Basidia** 21–29 × 6.5–10 μm, frequently bisporic to tetrasporic in all specimens, monosporic basidia observed frequently in L2 only and rarely trisporic found in BWN-34 and BWN-64, clavate or slightly truncate at apex, with abundant olivaceous granular content. **Cheilocystidia** abundant, polymorphous, short, broadly clavate, simple with terminal elements 9.5– 18.5 × 5.5–10 μm, while fusiform abundantly observed in L2 and L99 with terminal elements 22.5–26 × 7.5–9.5 μm, some having round apices, ante-terminal elements subglobose to cylindrical measuring 5–9.5 × 4.2–7 μm, with abundant olivaceous granular content. **Pleurocystidia** absent. **Pileipellis** constituted by cylindrical hyphae 2.5–11.5 μm in diam., frequently septate, sometimes branched, no constriction at septa, some hyaline others with internal brown vacuolar pigmentation, terminal elements with rounded ends. **Stipitipellis** constituted by hyphae 2–10.5 μm in diam., mostly cylindrical few broader, no or slight constriction at septa, hyaline in KOH, terminal elements with rounded tips. (Fig 7)

**Habit, habitat and distribution:** Growing solitary and in groups on sandy soil under *Acacia nilotica* trees in the desert and its surrounding areas. The distribution of this species is known from arid regions of Pakistan until now.

**Additional material examined:** PAKISTAN, Punjab, Lal Suhanra National Park, Cholistan desert, Bahawalpur, at 140 m a.s.l., scattered on sandy soil near *Acacia nilotica* trees, 30 August 2016, Hira Bashir, **L2**, (LAH35748), (GenBank # ON137218); Punjab, Lal Suhanra National Park, Cholistan desert, Bahawalpur, at 140 m a.s.l., scattered on sandy soil near *Acacia nilotica*, 7 August 2013, Hira Bashir and Muhammad Usman, **LS4**, (LAH35749), (GenBank # ON137219); Punjab, Toba Tek Singh, at 183 m a.s.l., solitary in grassy ground near *Dalbergia sissoo*, 30 June 2017, Hira Bashir, **TS56**, (LAH35750), (GenBank # ON137220). Punjab, District Bahawalpur, at 163 m a.s.l, from loamy soil along the canal bank under *Acacia nilotica*, 5 September 2020, Aneeqa Ghafoor and A.R. Niazi, **BWL 01** (LAH21719), (GenBank # ON158598). Punjab, District Bahawalnagar, at 163 m a.s.l, from loamy soil along the canal bank under *Acacia nilotica*, 31 July 2019, Muhammad Asif, **BWN 34** (LAH36808), (GenBank # ON158597). Punjab, District Bahawalnagar, at 163 m a.s.l, from loamy soil along the canal bank under *Acacia nilotica*, 11 August 2019, Muhammad Asif, **BWN 64** (LAH36807) (GenBank # ON158596). Punjab, District Bahawalnagar, at 163 m a.s.l, from nutrient rich loamy soil along the canal bank under *Acacia nilotica*, 11 August 2019, Muhammad Asif, **BWN 67**, (LAH36805), (GenBank # ON158600). Punjab, District Bahawalnagar, at 163 m a.s.l, from muddy soil along the canal bank under *Acacia nilotica*, 04 October 2019, Muhammad Asif, **BWN 85**, (LAH36806), (GenBank # ON158599).

**Notes**:

Phylogenetically, *A. dunensis* forms a small subclade with *A.* sp. (LAPAM35 and LAPAM64) and *A. parvibicolor*. These two species (LAPAM35 and LAPAM64) were introduced by Parra et al. 2018 [8] in the phylogenetic analyses and collected from Dominian Republic but not formally described yet. A few heteromorphic sites were observed in three samples (L2, L99 & TS56) of A. dunensis at differenet positions (Table 4). However, *A. parvibicolor* is characterized by its white and violaceous pileus with reddish or violet brown triangular fibrillose squamules while *A. dunensis* has pileus with bright yellow to orange yellow dotted squamules. Discoloration is yellow when stipe is cut and orange by handling in *A. parvibicolor*,

**Table 4. Polymorphic positions within different samples of *A. dunensis*.**

| Sample | Polymorphic positions/Nucleotide differences | | | | | | | |
|---|---|---|---|---|---|---|---|---|
| | 239 | 296 | 304 | 544 | 559 | 652 | 664 | 665 |
| BWL01 | G | G | T | T | T | G | - | - |
| BWN34 | G | G | T | T | T | G | - | - |
| BWN64 | G | G | T | T | T | G | - | - |
| L99 | R | G | T | T | Y | G | G | T |
| L2 | R | G | T | T | Y | G | G | T |
| LS4 | G | G | T | T | T | G | G | T |
| TS56 | A | R | - | Y | C | G | G | T |
| BWN67 | A | A | - | T | C | G | G | T |
| BWN85 | A | A | - | C | C | A | G | T |

Heteromorphisms are shown by yellow highlight. The symbol of (-) represents deletion.

in contrast, *A. dunensis* gave no significant discoloration when cut or very light yellow observed (L2) or faint yellowish brown at stipe base (BWN34 and BWN64). *Agaricus parvibicolor* has ellipsoid to elongate basidiospores having an average size of 6.3 × 3.9 μm [31] while *A. dunensis* has subglubose to ellipsoid basidiospres with an average size of 6.5 × 5.2 μm.

**Agaricus robustulus Linda J. Chen, Callac, L.A. Parra, K.D. Hyde & De Kesel 2017** (Figs 8A, 8B and 9)

**Material examined:** PAKISTAN, Punjab, University of the Punjab, Lahore, at 217 m a.s.l., solitary or in groups on rich loamy soil on the grassy grounds of the University Botanical Garden, 13 July 2016, Hira Bashir, **PU217** (LAH35764).

**Description: Pileus** 5–7 cm in diam., broadly convex to plano-convex, brown (3.8RP 4.6/2.9) squamules with reddish tinge dense at disc, sometimes disc becomes depressed (PU212), covered with irregular shaped, few roughly triangular or dotted squamules, sparse towards margin on creamy (3.1P 7.6/2.5) background. Surface dry and dull. **Margin** irregular, ruptured at maturity, slightly appendiculate, slightly incurved at maturity, vaguely exceeding lamellae, immediately become strongly fulvescent on handling or rubbing. **Lamellae** whitish (N10) to brown (3YR 1.3/3.8), free and approximate, crowded, intercalated with lamellulae and with entire edges. **Stipe** 3–7 × 0.5–2 cm, cylindrical having bulbous base, stuffed, provided with annulus in its upper half close to lamellae, above and below annulus covered with concolorous fibrils, strong yellow discoloration by rubbing or bruising, provided with multiple rhizomoprphs at the base. **Annulus** superous, single edged, membranous, fragile, smooth, white, non-persistent leaving behind ring zone on stipe at maturity. **Context** white, light yellow discoloration when cut predominantly from upper half of stipe. **Odor** of almond, mild. (Fig 8A and 8B)

**Macrochemical Reactions:** KOH reaction positive, yellow. Schäffer's reaction negative.

**Basidiospores** (5.5–) 5.9–6.3 (–6.6) × (3–) 3.5 (–4) μm, [avX = 5.8 ± 0.3 × 3.6 ± 0.2 μm, $Q_m$ = 1.5, n = 4 × 20], ellipsoid, rarely oblong, light to dark brown in KOH, smooth with a prominent apiculus. **Basidia** 14–22 × 6.5–9 μm, narrowly to broadly clavate, slightly truncate at apex, with abundant olivaceous granular content in KOH, frequently tetrasporic rarely bisporic. **Cheilocystidia** abundant, simple and septate at the base, with terminal elements 15–40 × 13.7–22.5 μm, polymorphous, broadly clavate, few pyriform, mostly hyaline in KOH, sometimes granular content present but not abundant, ante-terminal elements mostly cylindrical, broad, rarely subglobose, 3–10.5 × 2–6.9 μm. **Pleurocystidia** absent. **Pileipellis** constituted of interwoven hyphae 4–12.5 μm in diam., frequently septate, rarely branched, vaguely

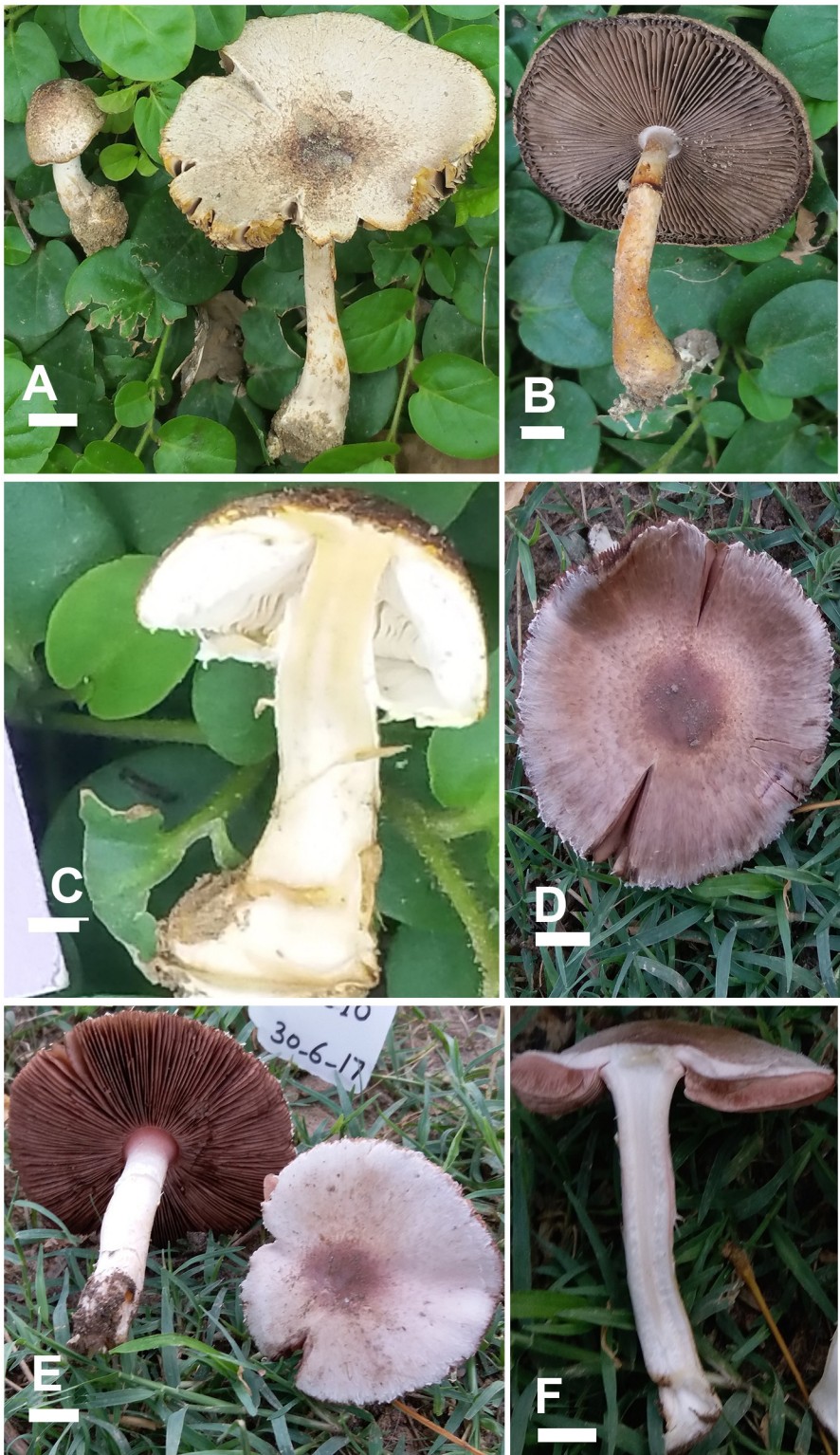

**Fig 8.** Macromorphological characters of *A. robustulus* (A-D) and *A. glabriusculus* (E-F). (A–F) basidiomata in the field. Bars = 1 cm. Photographed by Dr. Hira Bashir.

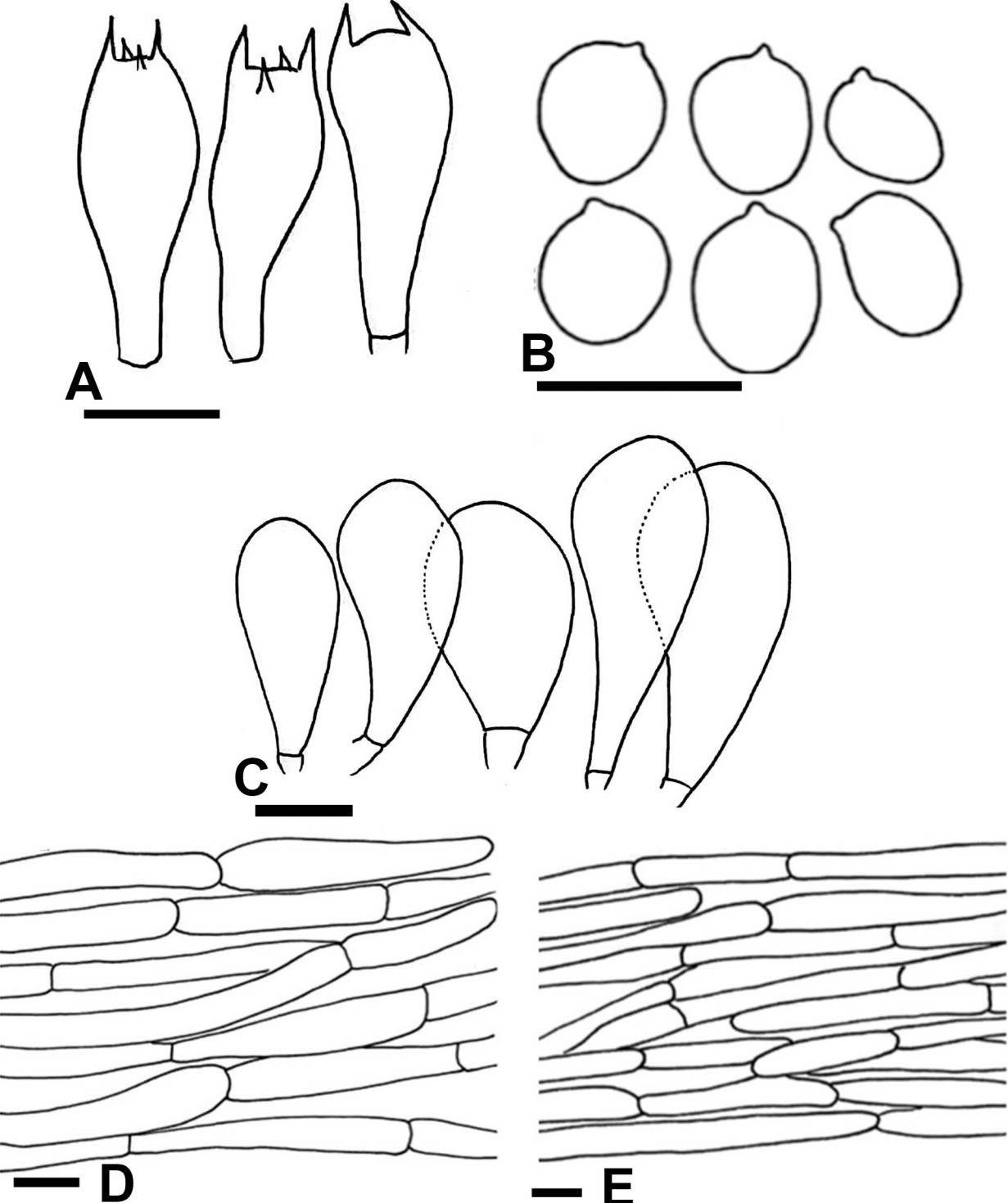

**Fig 9. Micromorphological characters of *A. robustulus* (A–E); (A) Basidia, (B) Basidiospores, (C) Cheilocystidia, (D) Pileipellis hyphae, (E) Stipitipellis hyphae.** Bars = 10 μm. Drawings by Dr. Hira Bashir.

constricted at septa, with internal brown vacuolar pigmentation or hyaline in KOH, terminal elements with more or less rounded ends, few having attenuated apices. **Stipitipellis** constituted by interwoven hyphae 2.5–9 µm in diam., cylindrical, septate, rarely branched, hyaline in KOH, slightly constricted at septa, with terminal elements having rounded ends. (Fig 9).

**Habit, habitat and distribution:** In groups, on grassy grounds of Botanical Garden, University of the Punjab and near *Eucalyptus* trees. This species is known from China and now from Pakistan so far.

**Additional material examined:** PAKISTAN, Punjab, University of the Punjab, Lahore, at 217 m a.s.l., solitary on rich loamy soil on the grounds, 13 July 2016, Hira Bashir, **PU212** (LAH35765); Punjab, University of the Punjab, Lahore, at 217 m a.s.l., solitary on rich loamy soil on the grounds of the University, 1 September 2016, Hira Bashir, **PU302** (LAH35766).

**Notes**:

Phylogenetically, four of our specimens get clustered with a recently described Chinese taxon, *A. robustulus*. This species is characterized by its reddish brown squamules on pileus and strongly fulvescent discoloration in mature sporocarps when rubbed or bruised. Cheilocystidia are polymorphic, having simple, broadly clavate to pyriform shaped. All these morphological characters are similar to that reported by Chen et al. 2017 [7]. Other species of this section also resemble morphologically to *A. brunneolus*, *A. goossensiae*, and *A. megalosporus*. However, the spore size in *A. goosensiae* is bigger (6.3 × 4.4) and cheilocystidia are inconspicuous while the other two taxa can be distinguished having large sized pileus exceeding 7 cm in diameter. This species is first time recorded from Pakistan.

*Agaricus glabriusculus* **S. Hussain** (Figs 8D–8F and 10)

**Diagnosis:** This species is distinguished well by having a pileus with brown fibrillose squamules dense at disc, context with pale brown discoloration when cut, annulus not very conspicuous, fragile, non-persistent and left in the form of ring zone on upper half of stipe. Basidia frequently bisporic to tetrasporic, cheilocysytidia broadly clavate and some have globose elements at apices.

**Description: Pileus** 4.5–6 cm in diam., broadly convex finally applanate, brown (3.8RP 4.6/2.9) squamules dense at disc, disc slightly depressed at maturity, few roughly triangular shaped, sparse towards margin at maturity on creamy background (3.1P 7.6/2.5), later surface turned light pinkish-brown. Surface fleshy, fragile. **Margin** serrate, ruptured at maturity, slightly appendiculate, not exceeding lamellae, light brown when rubbed. **Lamellae** pink (4.8RP 5.9/4.5) to brown (3YR 1.3/3.8), free and approximate, crowded, intercalated with lamellulae and with entire edges. **Stipe** 3.5–5 × 0.7–1 cm, having slightly bulbous base, cylindrical, stuffed, provided with annulus in its upper half close to lamellae, above and below covered with white fibrils, light pink and fleshy from top, solid, no significant discoloration or faint brown when rubbed, with rhizomorphs at the base in some speciemns. **Annulus** superous, single edged, membranous, fragile, smooth, white, non-persistent leaving behind ring zone on stipe. **Context** white, solid, no immediate discoloration when cut, faint brown color appears after 5–10 minutes. Odor faint almond-like. (Fig 8D–8F)

**Macrochemical Reactions:** KOH reaction positive, yellow. Schäffer's reaction negative.

**Basidiospores** (5.5–) 5.9–6.2 (–6.5) × (3.9–) <u>4.1</u> (–4.4) µm, [avX = 6.2 ± 0.44 × 4.2 ± 0.27 µm, $Q_m$ = 1.5, n = 20 × 3], ellipsoid, light to dark brown in KOH, smooth with a prominent apiculus. **Basidia** 20.5–24.2 × 6.8–8 µm, narrowly clavate, slightly truncate at apex, with abundant olivaceous granular content, frequently bisporic to tetrasporic. **Cheilocystidia** abundant, septate at the base, with terminal elements 11–16 × 6.5–9.5 µm, [avX = 13.3 ± 2.7 × 8 ± 1.4 µm, broadly clavate, sometimes having globose or rounded elements at apex, smooth, mostly hyaline in KOH, in some granular content observed but not abundantly, ante-terminal elements short cylindrical, broad, subglobose, 4–11 × 2.5–6 µm.

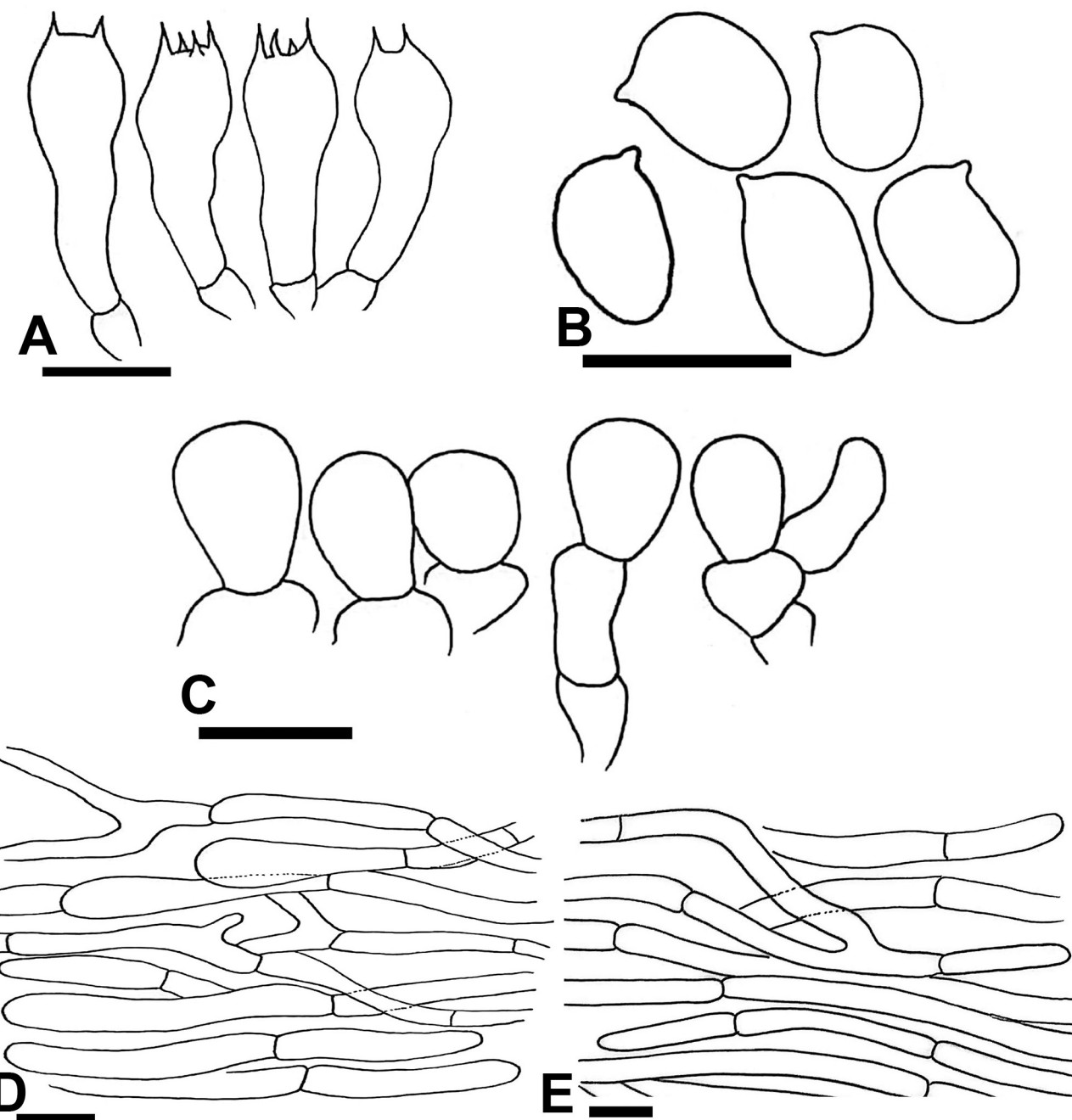

**Fig 10. Micromorphological characters of *A. glabriusculus* (A–E); (A) Basidia, (B) Basidiospores, (C) Cheilocystidia, (D) Pileipellis hyphae, (E) Stipitipellis hyphae.** Bars = 10 μm. Drawings by Dr. Hira Bashir.

**Pleurocystidia** absent. **Pileipellis** constituted by interwoven hyphae 4–12.5 μm in diam., frequently septate, frequently branched, constricted at septa, with internal brown vacuolar pigmentation or hyaline in KOH, terminal elements with rounded ends, clavate elements also observed. **Stipitipellis** constituted by interwoven hyphae 3–6 μm in diam., cylindrical, frequently septate, branched, hyaline in KOH, slightly constricted at septa, with terminal elements having rounded ends. (Fig 10)

**Habit, habitat and distribution:** In groups, on roadside in grassy soil and around wheat fields. It is known only from arid region of Pakistan so far.

**Material examined:** PAKISTAN, Punjab, Toba Tek Singh, at 183 m a.s.l., in groups on grassy ground along roadside, 30 June 2017, Hira Bashir, **TS10**, (LAH35761), (GenBank # ON158590). Punjab, Toba Tek Singh, at 183 m a.s.l., gregarious in grassy ground along road-side under bushes, 30 June 2017, Hira Bashir, **TS2**, (LAH35762), (GenBank # ON158591); Punjab, Toba Tek Singh, at 183 m a.s.l., in groups in a village about 29 km away from the city, on grassy ground near *Dalbergia sissoo* and *Populus* trees, 30 June 2017, Hira Bashir, **TS11**, (LAH35763), (GenBank # ON158592).

**Notes**:

*Agaricus glabriusculus* is an already reported species from Pakistan on the basis of ITS and LSU data [17] and India only on ITS data analysis [21]. In this study, Tef-1α of this species is also added. *Agaricus* sp. (MATA774), *A. purpureosquamulosus* and *A. goossensiae* are also close to *A. glabriusculus*. The polymorphic positions in different samples of *A. badiosquamulo-sus* and *A. glabriusculus* are given in Table 3. *Agaricus badiosquamulosus* is characterized by having dark brown with purplish tinged squamules, *A. glabriusculus* has fine pinkish fibrils on pileus surface at young stage later dark brown squamules covering the pileus thoroughly, dense at disc. In *A. badiosquamulosus*, strong yellowish-orange discoloration on pileus cover-ing and stipe when rubbed/touched and context becomes orange-brown or rusty upon bruis-ing or cutting, in *A. glabriusculus* no or faint brown discoloration appears upon cutting. *Agaricus* sp. (MATA774) has delicate, pinkish brown in young to dark brown pileus covering in mature sporocarps. Stipe white, light pink from the top above annulus with concolorous squamules and gives yellow discoloration when bruised.

*Agaricus violaceopunctatus* **H. Bashir** (Figs 11 and 12)

**Etymology:** '*violaceo*' originate from Latin meaning of violet color and '*punctatus*' means condensed at some point, which referred that our species has violet-colored squamules con-desnsed at disc of pileus.

**Holotype:** PAKISTAN, Punjab, University of the Punjab, Lahore, at 217 m a.s.l., solitary on rich loamy soil on the grassy grounds of the University, 11 July 2016, Hira Bashir, **PU125** (LAH35767).

**Diagnosis:** This species is well characterized by its violet squamules on pileus covering dense at disc and scattering towards margin at maturity, delicate and fleshy sporocarp, annulus fragile, white, and context with yellow discoloration on stipe when rubbed or bruised.

**Description: Pileus** 3–4.5 cm in diam., parabolic first, then convex, finally broadly convex at maturity, disc subumbonate in mature sporocarp, squamules predominantly provided with bright violet (8.9RP 3.9/5.3) coloration covering the pileus thoroughly at young stage, then sparse towards margin on creamy white background. Surface fleshy, fragile, becomes light yel-low (8.4YR 7.3/2.8) when rubbed. **Margins** entire in young sporocarp, radially ruptured at maturity, appendiculate, vaguely incurved, slightly exceeding lamellae. **Lamellae** whitish pink (0.9YR 7.1/2.9) at first, then light brown (6.6R 5.8/1), free and approximate, crowded, interca-lated with lamellulae and with entire edges. **Stipe** 4–7.5 × 1–2 cm, cylindrical with slightly bul-bous base, stuffed, provided with annulus in its upper half close to lamellae, above and below annulus covered with white appressed fibrils, white, pinkish abover annulus. **Annulus** super-ous, single edged, membranous, fragile, smooth, having white fibrils. **Context** white, discolor-ing light yellow on bruising. **Odor** of almonds, mild. (Fig 11)

**Macrochemical Reactions:** KOH reaction positive, yellow. Schäffer's reaction: positive, reddish orange on dry sporocarp.

**Basidiospores** 5–5.7 × 3.5–4.3 μm, [avX = 5.2 ± 0.23 × 3.8 ± 0.21, Qm = 1.39 ± 0.1, n = 2 × 20], ellipsoid, light brown in KOH, smooth with a prominent apiculus. **Basidia** 14–

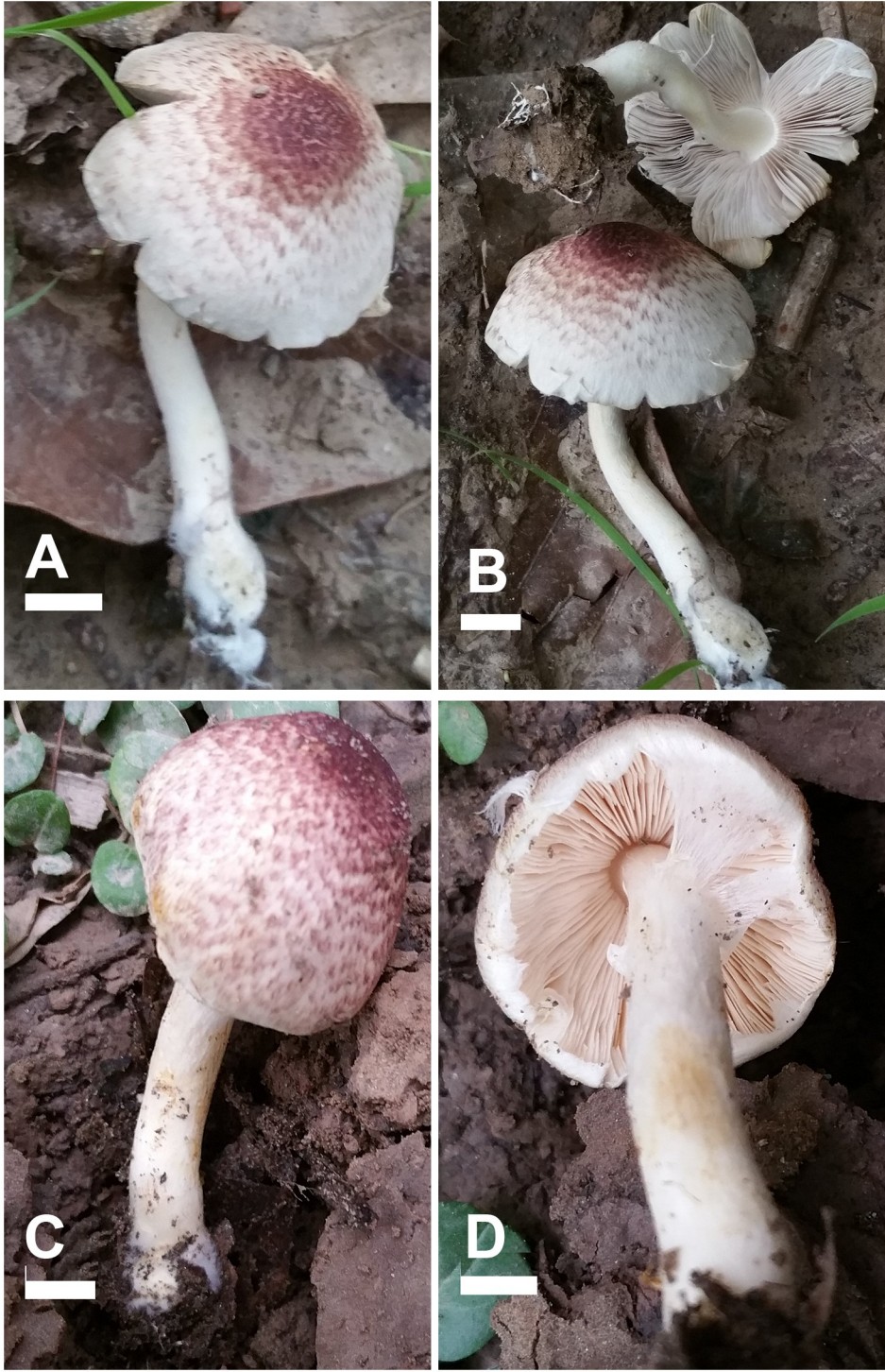

**Fig 11. Macromorphological characters of *A. violaceopunctatus* (A–D) basidiomata in the field.** Bars = 1 cm.
Photographed by Dr. Hira Bashir.

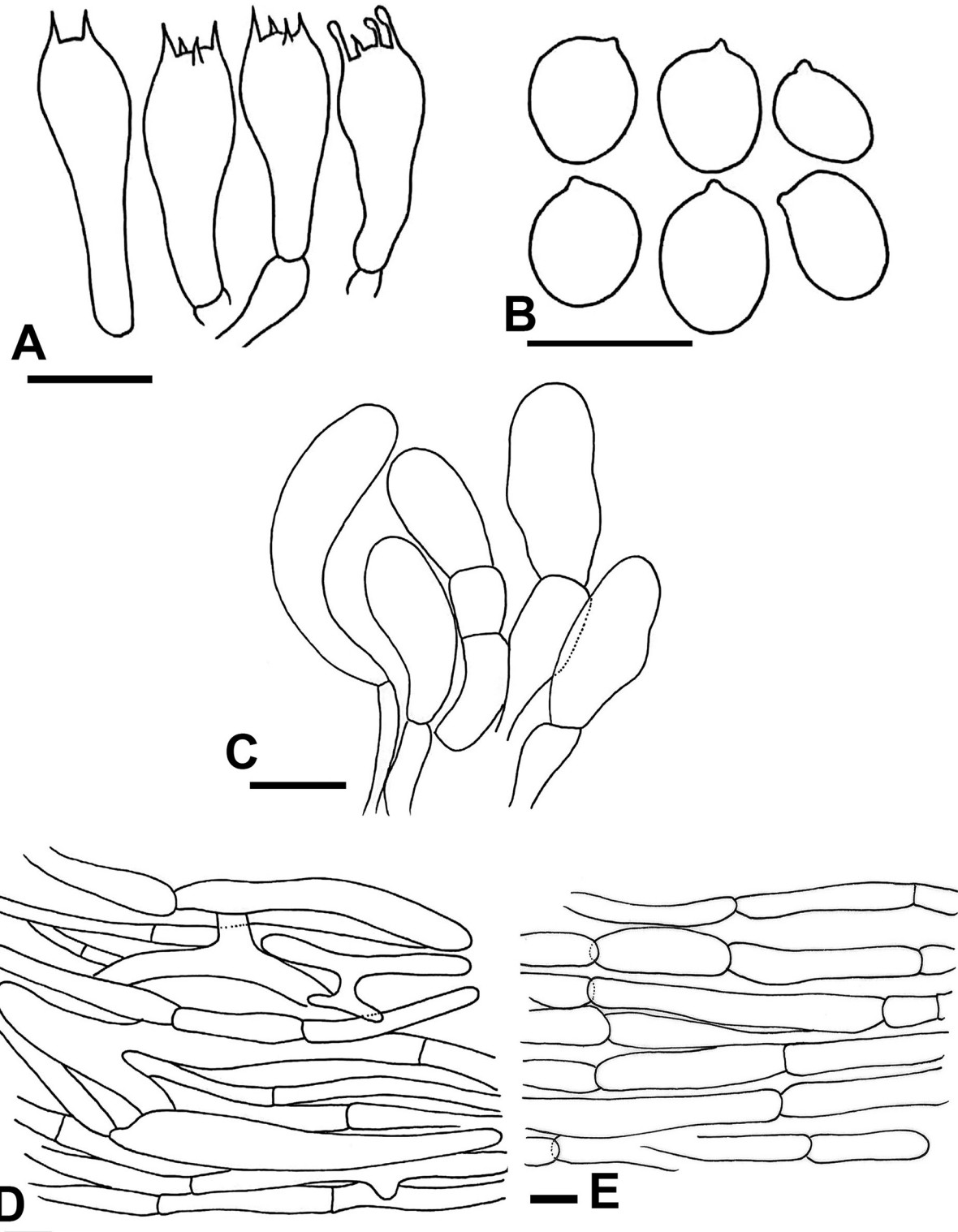

**Fig 12. Micromorphological characters of *A. violaceopunctatus* (A–E); (A) Basidia, (B) Basidiospores, (C) Cheilocystidia, (D) Pileipellis hyphae, (E) Stipitipellis hyphae.** Bars = 10 μm. Drawings by Dr. Hira Bashir.

25 × 6.2–7.5 µm, narrowly clavate, slightly truncate at apex, hyaline in KOH, frequently tetra-spoic rarely bisporic. **Cheilocystidia** simple or septate at the base, with terminal elements 15–30 × 6–10 µm, long and clavate, broadly clavate, ante-terminal elements mostly cylindrical, sometimes subglobose, hyaline in KOH. **Pleurocystidia** absent. Underside of annuluu. **Pilei-pellis** constituted by interwoven hyphae, 6–11 µm in diam., septate, branched, slightly constricted at septa, smooth, cylindrical, mostly with internal brown pigmentation and some hyaline in KOH, terminal elements with rounded ends. **Stipitipellis** constituted by hyphae 4–12.5 µm in diam., some broader hyphal elements observed interspersed in narrowly cylindrical hyphae, slightly constricted at septa, with terminal elements having rounded ends. (Fig 12).

**Habit, habitat and distribution:** In groups, on grassy grounds of Botanical Garden, University of the Punjab, near *Eucalyptus* trees. This species is known only from Lahore, Pakistan until now.

**Additional material examined:** PAKISTAN, Punjab, University of the Punjab, Lahore, at 217 m a.s.l., solitary on rich loamy soil on the grounds of the University, 13 July 2016, Hira Bashir, **PU234** (LAH35768). Punjab, Kasur disctrict, Changa Manga forest, scattered on ground, 218 m a.s.l., 21 July 2019, Aneeqa Ghaffor and A.R. Niazi, **CM16** (LAH21719).

**Notes**:

In phylogenetic analysis *A. violaceopunctatus* formed a subclade with all the undescribed taxa except *A. microviolaceous*. One of the closest taxon is the Pakistani specimen (MCR59), which was misidentified as *A. goossensiae* and the sequence is also submitted in GenBank (LAH5972011) with wrong identification. Our species has three nucleotide differences from *A.* sp. (MCR59) at position 37, 43 and 690 in ITS sequence data. When the original description of MCR59 (taken from the dissertation of author) was compared with our taxon, morphologically, pileus in MCR59 was recorded obtuse to umblicate with dark reddish squamules against reddish grey to hazel background and uplifted margin while in our species the squamules are strongly violaceous with creamy background. Annulus was not observed in MCR59 while in our collections annulus is very prominent, membranous and white. Clavate cheilocystidia were observed in MCR59 which are absent in our specimens. On the otherhand, *A. microviola-ceus* also has violet squamules, small sized sporocaps as observed in our newly described taxon but phylogenetically both species are separated having more than 15 nucleotide differences in their ITS sequence data. So, our new species does not similar to any of the already described taxa.

## Discussions

From Pakistan, three new species viz., *A. badiosquamulosus*, *A. dunensis*, and *A. violaceopunctatus* have been recorded during this study that are grouped within *A.* sect. *Minores*. In addition, one new record, *A. robustulus*, and an already reported species, *A. glabriusculus*, have also been documented providing their ITS, LSU, and Tef-1α sequence data. Two of those, *A. badiosquamulosus* and *A. glabriusculus*, clustered close to each other and are sister to *A.* sp. (MATA774) and with *A. goossensiae* (GF929; type specimen). *Agaricus* sp. (MATA774) was collected from a dune in 2007 growing near *Tulostoma* sp. by Philippe Callac & Gérardo Mata 2 Km away from Montepio, San Andrés Tuxtle, Mexico. *Agaricus* cf. *goossensiae* (ADK2751) was initially used in the phylogenetic analysis by Zhao *et al.* 2011 [5] and later as *A.* sp. [7, 9] so the species status was unclear. Therefore, in this study, we have requested to get a sequence of the type specimen of ADK2751 to Belgium herbarium which was provided to us for analysis. After comparisons, it was concluded that the ITS sequence of *A. goossensiae* (GF929) is far from the sequence of ADK2751 and both of these sequences do not share any genetic

character. Therefore, ADK2751 is considered to be another taxon, different from *A. goossensiae*. So, the later concept was taken into account and accepted. Similarly, another taxon from Pakistan was misidentified as *A. goossensiae* (MCR59; LAH5972011) retrieved from the GenBank that is revised as *A.* sp. in this study as the ITS sequence is quite different from the type of *A. goossensiae*. This species is, in fact, closely related to an undescribed species (ZRL2011039) from China. The same species from Pakistan *A.* sp. (MCR59) clustered close to *A. violaceopunctatus*, another new taxon reported in this study.

All of the four newly described species in this section were collected from subtropical semi-arid and arid regions of Pakistan. *Agaricus dunensis* was collected from Cholistan desert (L2, L99 and LS4) and forth collection (TS56) from Toba Tek Singh, from sandy soils having a dry arid climate. All the collections of *A. glabriusculus* were collected from Toba Tek Singh district having subtropical and arid climate. *Agaricus badiosquamulosus*, is a common species found abundantly in different areas of Lahore and one collection was made from Toba Tek Singh which are subtropical semi-arid and arid areas, respectively. *Agaricus robustulus* and *A. violaceopunctatus*, were collected from subtropical region of Lahore.

Mainly, the species of *A.* sect. *Minores* have been reported from China (Yunnan) and Thailand. From Thailand, the collections were made from tropical areas while the climate of Yunnan is subtropical. Previously, 27 species were reported from Thailand which are distributed in six clades and 14 species from Yunnan while only three species are shared by both of these regions [7]. After the detailed study on *A.* sect. *Minores* by Chen *et al.* 2017 [7], the section comprised of 93 species in total. He *et al.* 2017 [31] added sixteen new taxa in *A.* sect. *Minores*, from tropic origins, China. In this study, multigene and molecular dating analyses have been conducted and very interesting conclusions were made regarding the origin of *A.* sect. *Minores*. The study showed that this section has a tropical origin with four major routes: 1) ca. 9–13 Ma, species migrated to the Tibetan Plateau and Europe from South Asia; 2) ca. 22 Ma, species from outside South Asia dispersed to Europe; 3) around ca. 9 Ma, species spreaded to Alaska via North Asia, then West America from South Asia; and 4) species reached south and Oceania from South Asia by three invading occurrences around ca. 9, 12 and 16–18 Ma, respectively. He *et al.* 2018 [9] reported three new species of *A.* sect. *Minores* based on ITS and multigene phylogenetic data. Recently, Hussain and Sher added two new species in this section from Pakistan [17]. In the present study, 41 sequences are added of three newly described species including 29 ITS and six of LSU and Tef-1α regions. So, this section is very diverse having great number of species of tropic origin.

## Conclusion

The ITS and multigene phylogenetic analyses interpreted three new species *A. badiosquamulosus*, *A. dunensis*, and *A. violaceopunctatus*, a new record *A. robustulus* belonging to the *Agaricus* section *Minores* from subtropical regions of Pakistan. An already reported taxon from Pakistan and India, *A. glabriusculus* [17, 21] Tef-1α sequences are also added in this study which were not provided in the previous studies. All the micro- and macro-morphological data supported the molecular evidence. The species status of *Agaricus* cf. *goossensiae* (ADK2751) and *A. goossensiae* (MCR 59) on GenBank is clarified by obtaining the ITS sequence data from the type specimen of *A. goossensiae* (GF929) on special request and *Agaricus* sp. is assigned to both of these species. In this regard, the current investigation is an important contribution to the previously reported data on this largest section *Minores* of the genus *Agaricus*.

## Author Contributions

**Conceptualization:** Hira Bashir, Philippe Callac.

**Data curation:** Hira Bashir, Aneeqa Ghafoor, Abdul Rehman Niazi, Abdul Nasir Khalid, Gulnaz Parveen, Nidaa Harun, Najam-ul-Sehar Afshan, Ayesha Bibi, Philippe Callac.

**Formal analysis:** Hira Bashir, Philippe Callac.

**Investigation:** Hira Bashir, Muhammad Asif, Abdul Rehman Niazi, Abdul Nasir Khalid, Nidaa Harun, Philippe Callac.

**Methodology:** Hira Bashir, Abdul Nasir Khalid, Philippe Callac.

**Project administration:** Hira Bashir.

**Resources:** Hira Bashir, Muhammad Asif, Aneeqa Ghafoor, Abdul Rehman Niazi, Abdul Nasir Khalid, Gulnaz Parveen, Nidaa Harun, Philippe Callac.

**Software:** Muhammad Asif, Nidaa Harun, Philippe Callac.

**Supervision:** Abdul Rehman Niazi, Abdul Nasir Khalid, Philippe Callac.

**Validation:** Gulnaz Parveen, Najam-ul-Sehar Afshan, Philippe Callac.

**Visualization:** Gulnaz Parveen, Nidaa Harun, Najam-ul-Sehar Afshan, Ayesha Bibi, Philippe Callac.

**Writing – original draft:** Hira Bashir, Philippe Callac.

**Writing – review & editing:** Hira Bashir, Muhammad Asif, Philippe Callac.

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
