## [Decision Letter · Decision Letter 0]

7 Feb 2024

PONE-D-23-31131Multigene phylogeny and morphological descriptions of five species of Agaricus sect. Minores from subtropical climate zones of PakistanPLOS ONE

Dear Dr. Bashir,

Thank you for submitting your manuscript to PLOS ONE. After careful consideration, we feel that it has merit but does not fully meet PLOS ONE’s publication criteria as it currently stands. Therefore, we invite you to submit a revised version of the manuscript that addresses the points raised during the review process.

We look forward to receiving your revised manuscript.

Kind regards,

Dharmendra Kumar Meena

Academic Editor

PLOS ONE

A clean copy of the edited manuscript (uploaded as the new *manuscript* file)”.

4. We note that Figure 3, 5, 6, 8 and 11 in your submission contain copyrighted images. All PLOS content is published under the Creative Commons Attribution License (CC BY 4.0), which means that the manuscript, images, and Supporting Information files will be freely available online, and any third party is permitted to access, download, copy, distribute, and use these materials in any way, even commercially, with proper attribution. For more information, see our copyright guidelines: http://journals.plos.org/plosone/s/licenses-and-copyright.

a. You may seek permission from the original copyright holder of Figure 3, 5, 6, 8 and 11 to publish the content specifically under the CC BY 4.0 license. 

Additional Editor Comments:

Article is not acceptable in its present form . I invite you to revise in light of reviewers comments.

Reviewers' comments:

Reviewer's Responses to Questions

**Comments to the Author**

1. Is the manuscript technically sound, and do the data support the conclusions?

Reviewer #1: Yes

Reviewer #2: Partly

2. Has the statistical analysis been performed appropriately and rigorously? 

Reviewer #1: No

Reviewer #2: N/A

3. Have the authors made all data underlying the findings in their manuscript fully available?

Reviewer #1: No

Reviewer #2: Yes

4. Is the manuscript presented in an intelligible fashion and written in standard English?

Reviewer #1: No

Reviewer #2: Yes

5. Review Comments to the Author

Reviewer #1: Revise the manuscript and include more analysis to confirm the new species status. Many information should go to supplementary data. revise the language. rearrance the whole content. elaborate the comparision with other species.

Reviewer #2: Title of the paper: Multigene phylogeny and morphological descriptions of five species of Agaricus sect. Minores from subtropical climate zones of Pakistan

Abstract:

First two sentence of the abstract seems to be incomplete. Rewrite it.

Use the full form of ‘nuc’ in nuc rDNA.

What about the morphological description of the mentioned five species, you have only mentioned the geographical description. Kindly provide in brief about the molecular and morphological description of the mention five species.

At the end of the abstract put a sentences about the importance of the study and it will be useful.

Introduction: In Introduction part, although the authors has tried to explain in details about the mushroom species, the sentences are unclear. The aim/objectives of the study has not been mentioned.

Last sentence of the introduction should be in materials and method in subheading specimen collection and sampling sites

Materials and methods:

Subheading ‘Specimen collection ….’ The country name can be removed as it’s not required in the subheading.

Kindly explain the method of collected of the specimens and preservation/ fixation method for molecular analysis.

2nd sentence ‘based on morphological observations………’ should be in result part.

Kindly put coordinates for the locations and details of each location can be placed in proper way in the form of a table.

Subheading: Morphological observations- Correct the word ‘characeterization’

Mention the magnification of microscope used for measurement

Kindly provide the details of sequence submitted in MycoBank as a supplementary table.

Subheading ‘DNA sequence alignments and phylogenetic analyses’: correct the word ‘phylogeentic’ Mention the data analysis part in separate subheading

Results:

In table 1: Kindly mention the table heading

What does different color indicates in the table, kindly mention in notes.

‘The multigene sequence dataset…….. but still it remains……..data’: sentence in not clear. Modify it.

Why to put word notes there in result subheading. Give a proper subheading for the content.

Discussion:

‘Agaricus cf. goossensiae (ADK2751) was initially used in the phylogenetic analysis by Zhao et al. 2011 [29] later as A. sp.’-complete the sentence, its unclear.

Discussion part needs to be modified, citing more and recent references and comparing with the study.

No conclusion part. Kindly add the conclusion after the discussion

Figure of phylogeny is not at all clear. Replace it with high resolution figure.

Figures of the mushroom species have not been marked properly. Kindly recheck it.

6. PLOS authors have the option to publish the peer review history of their article (what does this mean?). If published, this will include your full peer review and any attached files.

Reviewer #1: No

Reviewer #2: No

---

## [Author Response · Author response to Decision Letter 0]

29 Feb 2024

All the authors appreciate and respect the remarks and comments of both reviewers. All the corrections are inserted in the text file as per instructions. All the changes are highlighted in yellow in the text file as well as comments are addressed below one by one (yellow highlighted sections). We are thankful to both of the reviewers especially reviewer 2 for the constructive suggestions and corrections.

For the Ethical statement, it is to note that our ethical committee permits to collect of samples or data if it is related to animal or human-based material/specimens. For plants or fungal specimens, an ethical statement is not required. Therefore, the ethical statement does not apply to our research work. If the Reviewers still have not agreed, the ethical statement can be obtained upon special request of the authors. It is requested to please proceed with our manuscript without the ethical statement as it is not applicable here.

Note: This point is discussed with the academic editor, Plos One, and the editor is agreed if reviewers have no objection.

Reviewer #1: Revise the manuscript and include more analysis to confirm the new species status. Many information should go to supplementary data. revise the language. rearrance the whole content. elaborate the comparision with other species.

Response: Your comments are respectable to us. I hereby confirm that in taxonomy based studies (you can compare with previously published work) that morphological, anatomical, and molecular data based on ITS and Multigene (ITS+LSU+Tef-1 alpha) regions is required for publication of genus Agaricus. Dr. Philippe Callac (Late) an expert of this genus has approved that these analyses are enough to publish this data. In fact, several publications lack LSU and Tef sequence data which is completely provided for all the new species to confirm their unique status and to report them as new species in our studies. It is therefore requested to accept this manuscript with already provided detailed morpho-anatomical descriptions, ITS and multigene based molecular analyses, and their comparisons with closely related taxa one by one in subheading of “notes” provided after descriptions in each species. 

Reviewer #2: Title of the paper: Multigene phylogeny and morphological descriptions of five species of Agaricus sect. Minores from subtropical climate zones of Pakistan.

Abstract:

--First two sentence of the abstract seems to be incomplete. Rewrite it.

Response: First two sentences are rewritten.

--Use the full form of ‘nuc’ in nuc rDNA.

Response: Done.

--What about the morphological description of the mentioned five species, you have only mentioned the geographical description. Kindly provide in brief about the molecular and morphological description of the mention five species.

Response: At the end of the abstract, it is mentioned that all species are described on the basis of morphological and molecular data (DNA regions are already mentioned). The detailed morphological characteristics are mentioned in the results section.

--At the end of the abstract put a sentences about the importance of the study and it will be useful.

Response: Done

--Introduction: In Introduction part, although the authors has tried to explain in details about the mushroom species, the sentences are unclear. The aim/objectives of the study has not been mentioned.

Response: Purpose of study is mentioned at the end of the introduction.

--Last sentence of the introduction should be in materials and method in subheading specimen collection and sampling sites. 

Response: Done

Materials and methods:

--Subheading ‘Specimen collection ….’ The country name can be removed as it’s not required in the subheading.

Response: Done 

--Kindly explain the method of collected of the specimens and preservation/ fixation method for molecular analysis.

Response: Please see morphological characterization, all the methodologies are described in detail.

--2nd sentence ‘based on morphological observations………’ should be in result part.

Response: Done

--Subheading: Morphological observations- Correct the word ‘characeterization’

Response: done 

--Mention the magnification of microscope used for measurement.

Response: Done.

--Kindly provide the details of sequence submitted in MycoBank as a supplementary table.

Response: All the sequences are submitted to the GenBank and table is already provided indicating accession numbers (ITS, LSU, Tef1 regions).

--Subheading ‘DNA sequence alignments and phylogenetic analyses’: correct the word ‘phylogeentic’ Mention the data analysis part in separate subheading

Response: Done.

Results:

--In table 1: Kindly mention the table heading

Response: Already mentioned as below: 

Table 1. All the representatives of Agaricus sect. Minores sequences of known and unnamed species are mentioned along with the outgroup and other sections samples.

--What does different color indicates in the table, kindly mention in notes.

Response: Added in the text also.

Table 1. All the representatives of Agaricus sect. Minores sequences of known and unnamed species are mentioned along with the outgroup and other sections samples.

(A) the newly named species since 2000 have their names ajusted to the left in the column (names in black letters for new species and green letters for renamed species) 

(B) species named before 2000 are adjusted to the right of the column and are in red letters

(C) the unnamed or provisionally named species (cf. or aff.) are on the left and in blue letters

(D) For each taxon or putative taxon a single sample, the Type specimen T if available, is shown but we have made some exceptions and included more than one sequence

---for samples that are not the type but which have more sequence data (some are paratype P but we have not verified all)

---for samples of the same species later reported from distant countries (generally different continents indicted with initial AF, AS, EU, NA, SA, OC, and CA for the Caribbean)

--‘The multigene sequence dataset…….. but still it remains……..data’: sentence in not clear. Modify it.

Response: Done.

--Why to put word notes there in result subheading. Give a proper subheading for the content.

Response: These are not the contents, in fact comparative notes where all the cpecies are compared with the closely related taxa morphologically as well as moleculary. I request to keep it same.

Discussion:

--‘Agaricus cf. goossensiae (ADK2751) was initially used in the phylogenetic analysis by Zhao et al. 2011 [29] later as A. sp.’-complete the sentence, its unclear.

Response: Done. 

-- Discussion part needs to be modified, citing more and recent references and comparing with the study.

Response: Actually all the comparisons with related species have been made in “notes” subheading of each species. If I add the comparisons here also, it will be the repetition. Therefore it is requested to keep this section the same to avoid repetitions.

--No conclusion part. Kindly add the conclusion after the discussion.

Response: Added.

--Figure of phylogeny is not at all clear. Replace it with high resolution figure.

Figures of the mushroom species have not been marked properly. Kindly recheck it.

Response: Already high resolution TIFF images are provided. The trees are too long specially ITS, so it it is requested from the journal to split the tree on 2 pages so that it can be readable to readers.

---

## [Decision Letter · Decision Letter 1]

1 Apr 2024

Multigene phylogeny and morphological descriptions of five species of Agaricus sect. Minores from subtropical climate zones of Pakistan

PONE-D-23-31131R1

Dear Dr. Basir

We’re pleased to inform you that your manuscript has been judged scientifically suitable for publication and will be formally accepted for publication once it meets all outstanding technical requirements.

Kind regards,

Dharmendra Kumar Meena

Academic Editor

PLOS ONE

Additional Editor Comments (optional):

article can be accepted

Reviewers' comments:

Reviewer's Responses to Questions

**Comments to the Author**

1. If the authors have adequately addressed your comments raised in a previous round of review and you feel that this manuscript is now acceptable for publication, you may indicate that here to bypass the “Comments to the Author” section, enter your conflict of interest statement in the “Confidential to Editor” section, and submit your "Accept" recommendation.

Reviewer #2: All comments have been addressed

2. Is the manuscript technically sound, and do the data support the conclusions?

Reviewer #2: Yes

3. Has the statistical analysis been performed appropriately and rigorously? 

Reviewer #2: Yes

4. Have the authors made all data underlying the findings in their manuscript fully available?

Reviewer #2: Yes

5. Is the manuscript presented in an intelligible fashion and written in standard English?

Reviewer #2: Yes

6. Review Comments to the Author

Reviewer #2: The authors has inculded and addressed the comments/suggestion given to them for correction in the paper. Therefore, the research paper can be accepted for publication.

7. PLOS authors have the option to publish the peer review history of their article (what does this mean?). If published, this will include your full peer review and any attached files.

Reviewer #2: No
